# A multidisciplinary approach to a unique palaeolithic human ichnological record from Italy (Bàsura Cave)

Marco Romano[1]*, Paolo Citton[2], Isabella Salvador[3], Daniele Arobba[4], Ivano Rellini[5], Marco Firpo[5], Fabio Negrino[6], Marta Zunino[7], Elisabetta Starnini[8,9], Marco Avanzini[3]

[1]Evolutionary Studies Institute (ESI), School of Geosciences, University of the Witwatersrand, Johannesburg, South Africa; [2]CONICET-Consejo Nacional de Investigaciones Científicas y Técnicas, Buenos Aires, Argentina; [3]MUSE, Museo delle Scienze, Trento, Italy; [4]Museo Archeologico del Finale, Finale Ligure Borgo, Italy; [5]Dipartimento di Scienze della Terra, dell'Ambiente e della Vita (DISTAV), Università degli Studi di Genova, Genoa, Italy; [6]Dipartimento di Antichità, Filosofia, Storia (DAFIST), Università degli Studi di Genova, Genoa, Italy; [7]Grotte di Toirano, Toirano, Italy; [8]Soprintendenza Archeologia Belle Arti e Paesaggio per la Città Metropolitana di Genova e le province di Imperia, La Spezia e Savona, Genoa, Italy; [9]Dipartimento di Civiltà e Forme del Sapere, Università di Pisa, Pise, Italy

**Abstract** Based on the integration of laser scans, sedimentology, geochemistry, archeobotany, geometric morphometrics and photogrammetry, here we present evidence testifying that a Palaeolithic group of people explored a deep cave in northern Italy about 14 ky cal. BP. Ichnological data enable us to shed light on individual and group level behavior, social relationship, and mode of exploration of the uneven terrain. Five individuals, two adults, an adolescent and two children, entered the cave barefoot and illuminated the way with a bunch of wooden sticks. Traces of crawling locomotion are documented for the first time in the global human ichnological record. Anatomical details recognizable in the crawling traces show that no clothing was present between limbs and the trampled sediments. Our study demonstrates that very young children (the youngest about 3 years old) were active members of the Upper Palaeolithic populations, even in apparently dangerous and social activities.
DOI: https://doi.org/10.7554/eLife.45204.001

*For correspondence:
marco.romano@uniroma1.it

Competing interests: The authors declare that no competing interests exist.

## Introduction

Discovered in 1950, the hypogeal part of the 'Grotta della Bàsura' is a large and deep cave that has produced some of the most important Italian Palaeolithic discoveries of the twentieth century (*Chiappella, 1952*; *Tongiorgi and Lamboglia, 1954*; *Blanc, 1960*; *Lamboglia, 1960*; *Giacobini, 2008*) consisting of traces of human activity, especially footprints, dating back to 14 ky cal. BP. Up till 1890 only the atrial part of the cavity was known where Neolithic and late Roman archaeological finds had been unearthed (*Maineri, 1985*). The cave opening is situated at 186 m a.s.l. about 1 km north of Toirano (Savona, Italy - 436253.433 E; 4887689.739 N), and extends 890 m into Mount S. Pietro with an elevation difference of +20/−22 m relative to that of the entrance (*Figure 1*).

The inner rooms became accessible in 1950, after the rupture of a stalagmite column, placed a few dozen meters from the entrance, that prevented any access to the cave (*Tongiorgi and Lamboglia, 1954*; *Blanc, 1960*; *Lamboglia, 1960*). The exceptional prehistoric and paleontological value of

**eLife digest** The fossil traces of Stone Age humans and other animals in the Grotta della Bàsura cave system in Italy have been studied since the 1950s. Italian archaeologist Virginia Chiappella published the first studies; she documented bones from an extinct cave bear, human and animal footprints, charcoal from torches, finger marks, and lumps of clay stuck on the walls. Since then, many more archeologists and anthropologists have studied the cave and its fossils. Yet there are still lessons to be learned from this prehistoric site.

Now, Romano et al. have combined a number of different approaches and used some of the latest technology and cutting-edge software to analyze 180 footprints and other tracks found in the cave. These trace fossils date to about 14,000 years ago, and the analysis revealed that they were left by a group of Stone Age humans who descended at least 400 meters into the cave. The group consisted of two adults, an adolescent and two children of about three and six years old. At one point they had to crawl through a low tunnel – something that has not previously been documented in the fossil record. The group were all barefoot, had no clothing on their arms and legs and used wooden torches to light the way.

Together, these findings suggest that young children were active group members during the late Stone Age, even when carrying out apparently dangerous activities. Romano et al. now hope that their multidisciplinary approach may help other scientists looking to understand how humans behaved elsewhere in the world at various points in history.

DOI: https://doi.org/10.7554/eLife.45204.002

the cave was firstly recognized by Virginia Chiappella (*Chiappella, 1952*) who was the first scholar to visit the site soon after its discovery. Chiappella identified several bones of *Ursus spelaeus* and traces both of animal and human frequentation (footprints, charcoals, digital tracks, lumps of clay adhering to the walls) in different areas of the cave within approximately 350 m of the entrance. Regrettably, destruction of most of the ichnological record occurred as a result of uncontrolled cave visits by local villagers and tourists lured to the remarkable discovery as a result of media reports (*Blanc et al., 1960*; *De Lumley and Giacobini, 1985*).

The first study of human footprints from the Cave of Bàsura was conducted by *Pales (1960)*, based on original images and 13 plaster casts of the best-preserved specimens from various sectors of the cave. Analysis of the bone architecture of the footprint makers and the apparent relationship with remains of *Ursus spelaeus*, led the author to consider that the footprints were made by 'Neanderthal-type' producers. Subsequent analysis (*de Lumley and Vicino, 1984*), accompanied by radiometric dating, placed the prehistoric visitation of the Bàsura in the Upper Palaeolithic, between 12,000 and 14,000 years uncal BP (*de Lumley and Vicino, 1984*; *De Lumley and Giacobini, 1985*). Dating of charred wood fragments found on the trampled surface provided a more precise age for cave visitation at 12,340 years ± 160 years BP (*Molleson et al., 1972*; *Molleson, 1985*).

Based on the ichnological study of Pales, Blanc proposed that the inner room called 'Sala dei Misteri' was reached by multiple individuals, including a juvenile. *Blanc (1960)* described, from the same room, a group of seven human footprints identified by heel tracks positioned only a few centimeters from the main wall of the chamber to which numerous lumps of clay were attached. The close association of the footprints with lumps of clay resulted in it being considered the result of initiation rites involving young hunters. This hypothesis was supported by the presence of a stalagmite concretion (defined by Blanc as '*acephalous sphinx*' or '*zoomorphic stalagmite*') placed against the terminal wall of the 'Sala dei Misteri' (Mysteries' Hall) chamber on which several sinuous furrows had been made intentionally by several individuals using their fingers. This interpretation is currently under review by two of the present authors (ES, MZ). The aim of this work is to review and improve understanding of the Grotta della Bàsura's human ichnology, soil micromorphology, sedimentology and radiocarbon chronology. A preliminary report from this study focused on human footprints preserved in the innermost chamber of the cave (*Citton et al., 2017*).

In the present paper, we expanded our study to analyze and interpret all human traces (footprints and handprints as well as other traces) from the 'Grotta della Bàsura', providing new insight on the

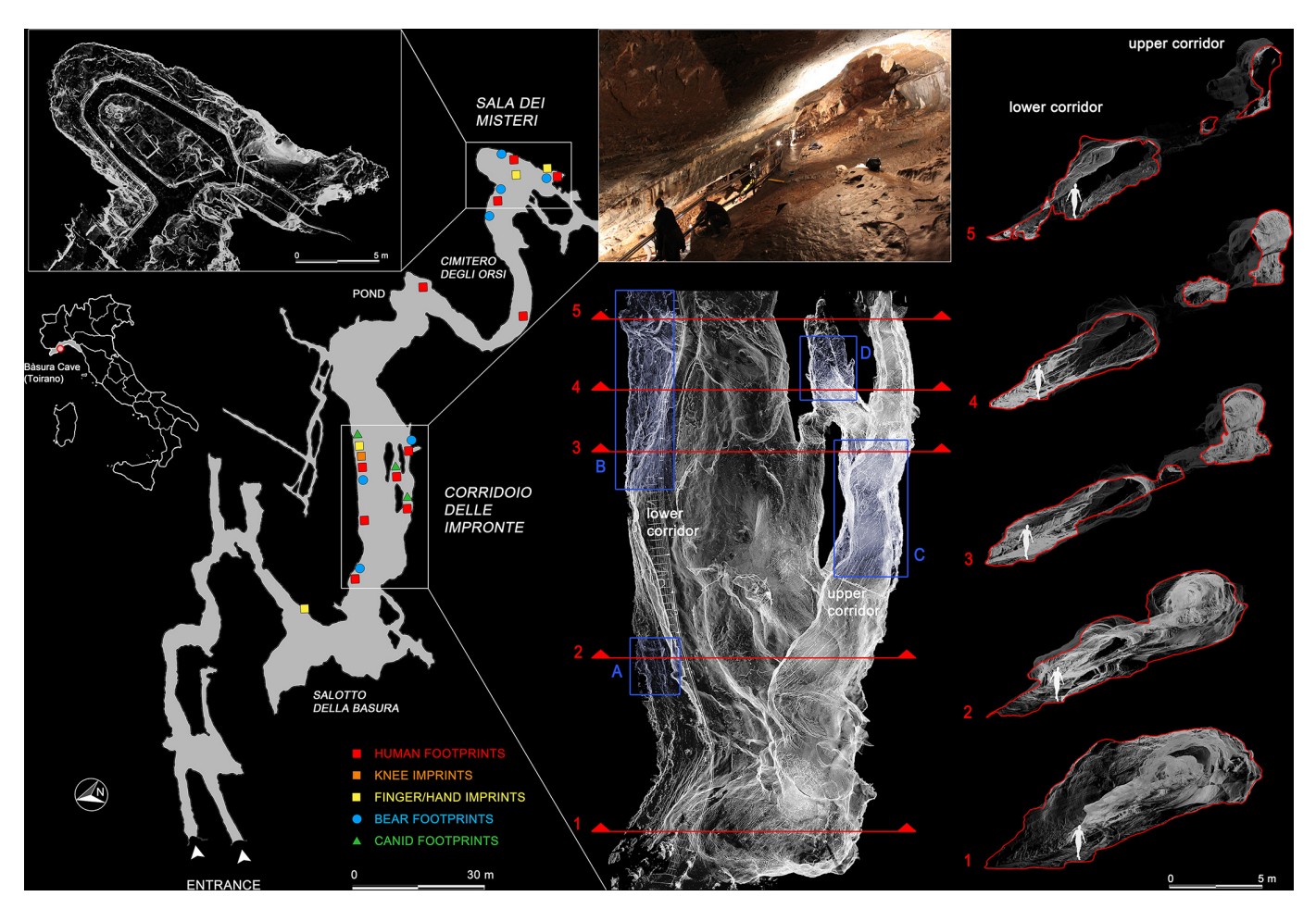

**Figure 1.** Planimetry of the '*Grotta della Bàsura*' and location of human, bear and canid footprints. White rectangles enclose the three-dimensional reconstructions, obtained via laser scanner, of the innermost room ('*Sala dei Misteri*' - left) and the main gallery ('*Corridoio delle impronte*' - right) of the cave, where the human footprints are preserved. Cross-sections obtained from the three-dimensional reconstruction of the main gallery are highlighted in red and show the branching of the 'lower' and 'upper' corridors, respectively. Blue rectangle indicate the four areas within the main gallery where most of the human footprints are concentrated (A and B for the lower corridor, C and D for the upper corridor).
DOI: https://doi.org/10.7554/eLife.45204.003

behavior, identity of the members, their exploratory techniques, and the social structure of an Upper Palaeolithic group.

## Data

A total of 180 footprints and traces *sensu lato* were recorded and studied (*Supplementary file 1*). In addition to footprints (*Figure 2*), among the traces are digit and handprints on the clay-rich floor, and smears from hands dirtied with charcoal on the side walls of the cave (*Figure 3*) (*Giannotti, 2008*). A paleo-archaeological excavation was performed in 2016 in the '*Sala dei Misteri*'. This survey highlighted the total absence of archaeological material but led to the recovery, on the trampled palaeosurface, of numerous charcoal remains with bundles of *Pinus* t. *sylvestris/mugo* originally used to illuminate the cave. New radiometric dating on these charcoal samples constrains the exploration of the cave to the late Upper Palaeolithic, between 12,310 ± 60 and 12,370 ± 60 BP, that is from about 14,700 to 14,000 cal BP (*Table 1*).

Digit and hand traces are preserved in several sectors of the cave (*Figure 3*). Most are unintentional traces related to cave exploration activities (*Figure 3*, C0, C26b, C72). Others, especially in the inner chamber ('*Sala dei Misteri*') which are still being studied, are very probably related to social

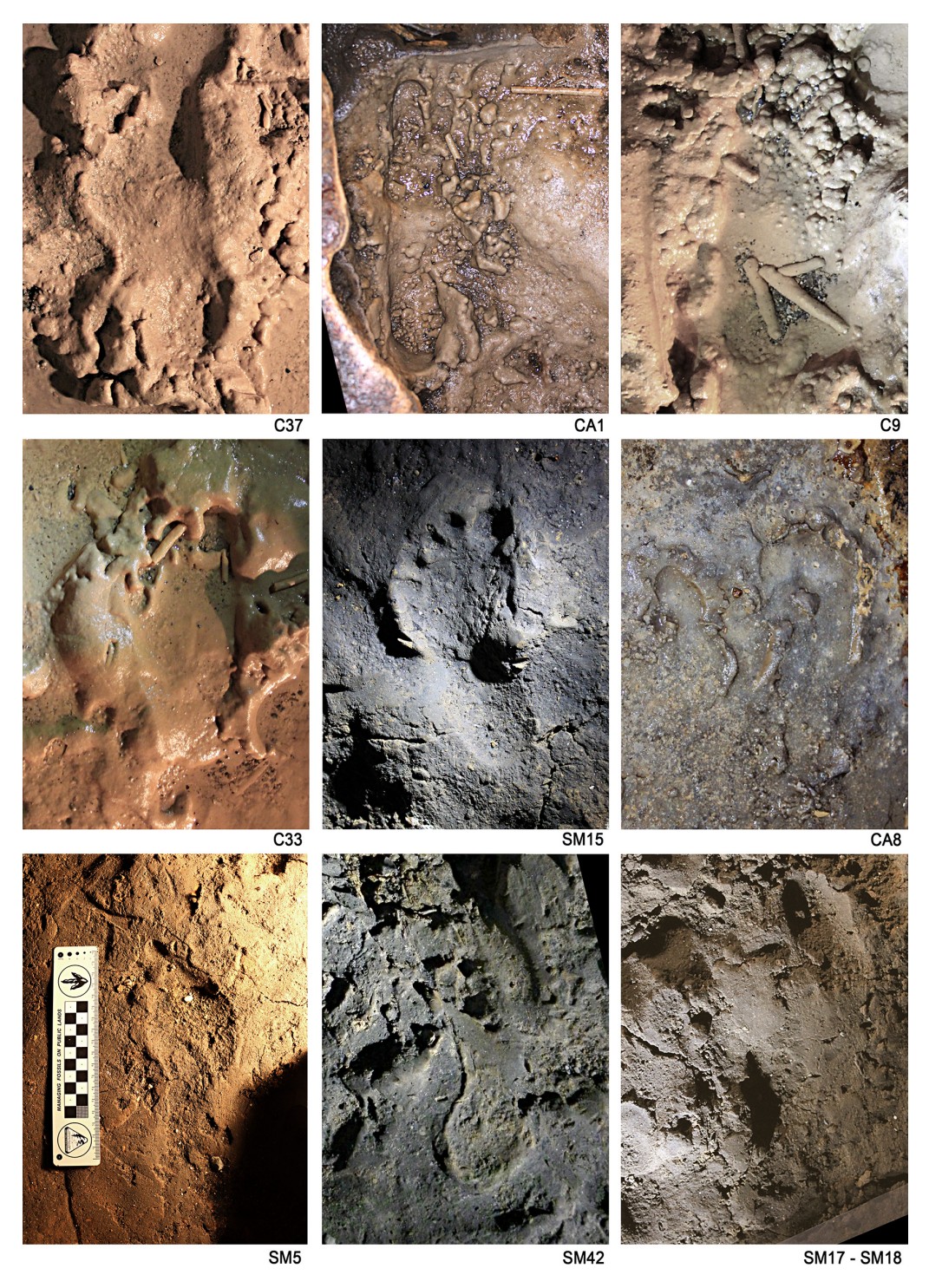

**Figure 2.** Human footprints imprinted on muddy substrate in different moisture conditions. C37, Human footprint referred to the Morph. 5 ('lower corridor'). CA1 and C9, Human footprint referred to the Morph. 4 ('upper corridor'). C33, Human footprint referred to the Morph. 3 ('lower corridor'). SM15, Human footprint referred to the Morph. 3 ('Sala dei Misteri'). CA8, Human footprint referred to the Morph. 3 ('upper corridor'). SM5 and SM42, Human footprint referred to the Morph. 2 ('Sala dei Misteri'). SM17 and SM18, Human footprint referred to the Morph. 1 ('Sala dei Misteri').

DOI: https://doi.org/10.7554/eLife.45204.004

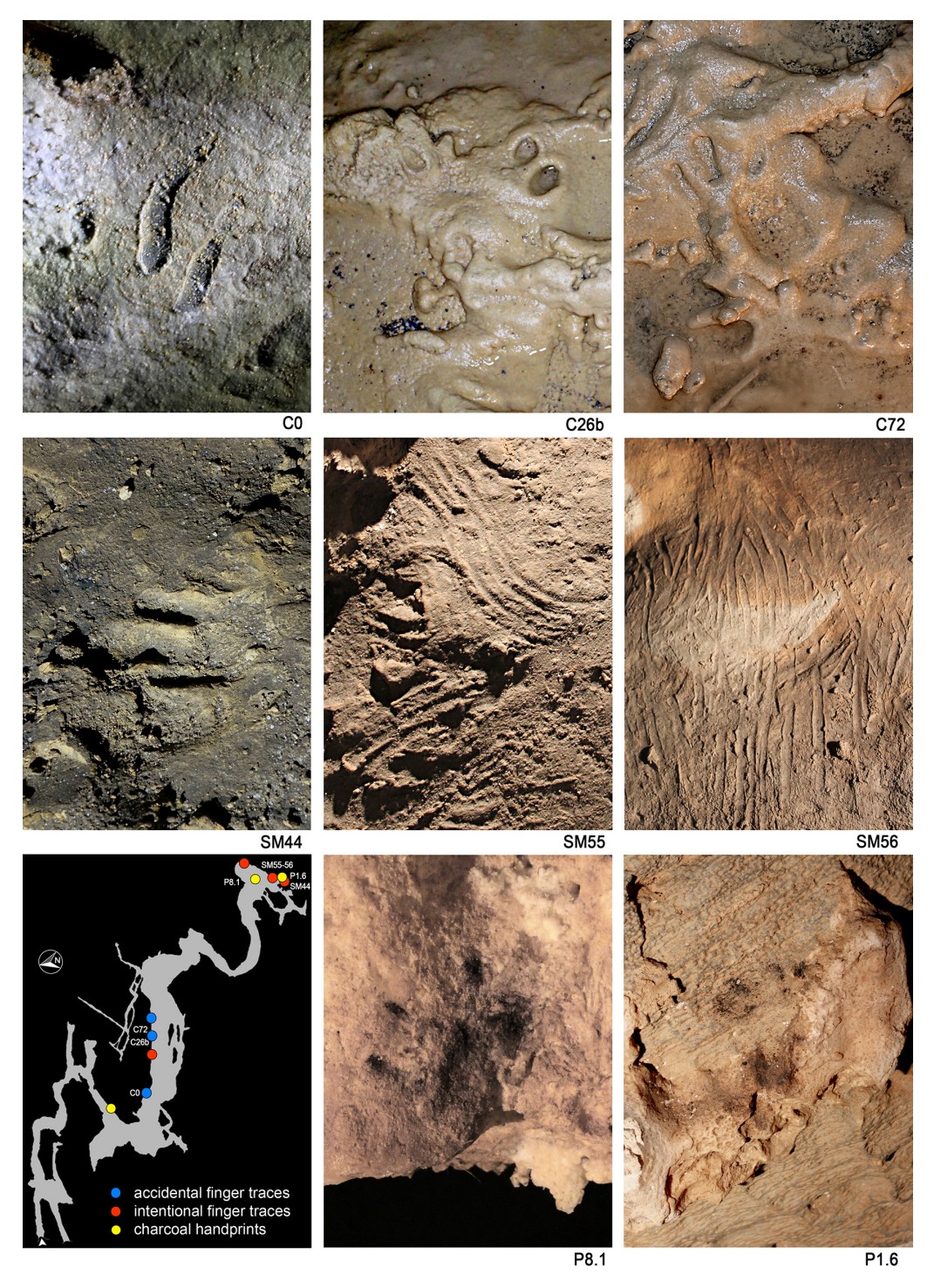

**Figure 3.** Finger and hand prints. C0, Two finger traces on the concretioned side-wall of the 'lower corridor'. C26b, Finger traces ('lower corridor'). C72, Hand print ('lower corridor'). SM44, finger traces ('Sala dei Misteri'). SM55, Finger flutings on the clay on the clay floor ('Sala dei Misteri'). SM56, Finger flutings on a clay-coated stalagmite ('Sala dei Misteri'). P8.1, P1.6, Coal dirtied handprints ('Sala dei Misteri').
DOI: https://doi.org/10.7554/eLife.45204.005

**Table 1.** Radiometric dating of charcoals collected from the trampling palaeosurface during 2017 excavations inside the 'Sala dei misteri'. *14C ages have been calibrated to calendar years with software program: OxCal, version 4.3. Used calibration curve: IntCal13.

| Sample name | Provenance | Dated material | Description | Lab. code | F$^{14}$C ± 1σ | $^{14}$C Age (yr BP) ±1σ | Calibrated* dating result (95.4 probability) | %C | %N | δ$^{13}$C (in ‰;±1σ) | δ$^{15}$N (in ‰;±1σ) | C/N ratio |
|---|---|---|---|---|---|---|---|---|---|---|---|---|
| Bàsura SM B5 17B | 'sala dei misteri', square B5, Unit 1 | Charcoal (AAA) | Pinus t. sylvestris /mugo | GrA-69598 | 0.2160 ± 0.0016 | 12310 ± 60 | 12720–12110 cal BC | 71.4 | - | -25.24 ± 0.14 | - | - |
| Bàsura SM D6 33B | 'Sala dei misteri', square D6, Unit 1 | charcoal (AAA) | Pinus t. sylvestris /mugo | GrA-69597 | 0.2145 ± 0.0016 | 12370 ± 60 | 12830–12165 cal BC | 63.3 | - | -26.37 ± 0.14 | - | - |

DOI: https://doi.org/10.7554/eLife.45204.006

or symbolic activities and can be instead considered intentional (*Figure 3*, SM55, SM56). Moreover, different sized bear and Canidae *incertae sedis* footprints are ubiquitously present (*Figure 4*), and are often associated with human prints. The available data regarding the footprints attributed to 'canids' suggest a very reduced number of individuals and a close association with the human prints. Should ongoing studies (Avanzini et al in prep.) confirm that the ichnnological association of Bàsura could prove crucial to shed new light on dog domestication in the Upper Paleolithic (*Morey and Jeger, 2015*; *Perri, 2016*; *Lupo, 2017*; *Janssens et al., 2018*).

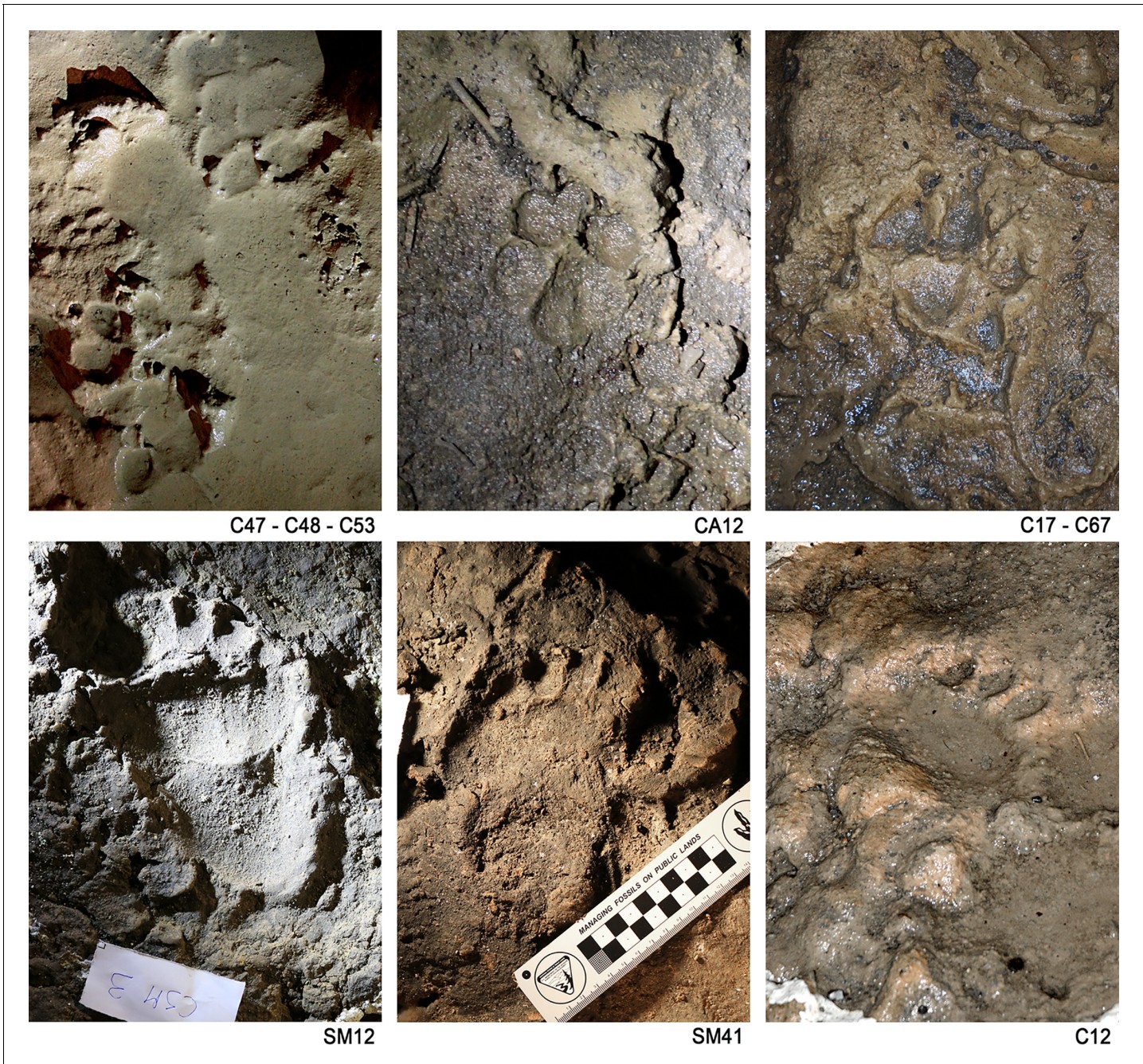

**Figure 4.** Canidae *incertae sedis* and bear footprints. C47-C48-C53 Canidae footprint on saturated mud ('upper' corridor). CA12 well preserved Canidae footprint ('upper corridor'). SM12-SM41 bear footprint (Sala dei Misteri). C12 bear handprint ('lower corridor').
DOI: https://doi.org/10.7554/eLife.45204.007

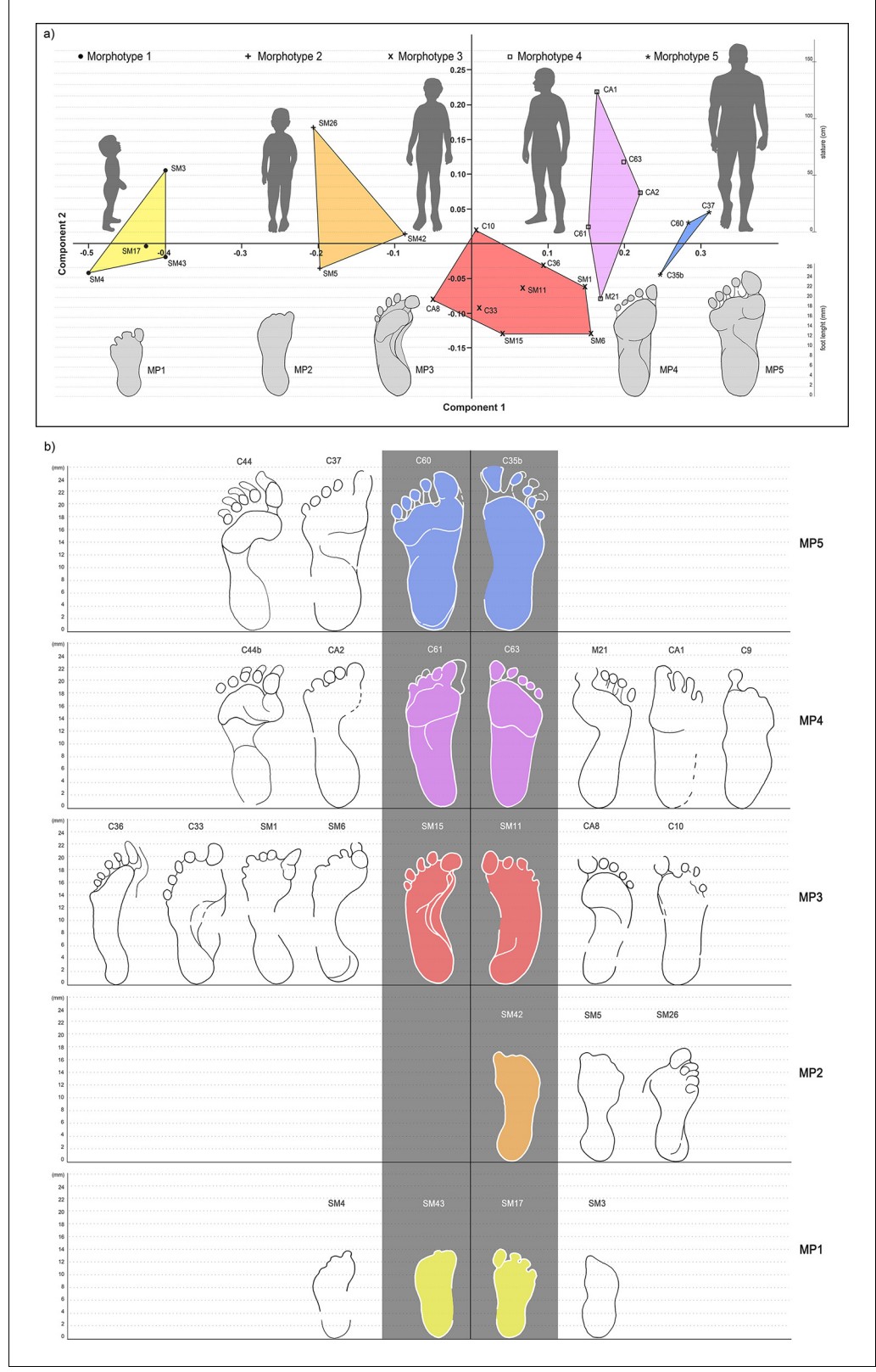

**Figure 5.** Principal Component Analysis based on the best-preserved footprints from *'Sala dei Misteri'* and *'Corridoio delle impronte'*. (a) The five morphotypes to which footprints have been referred are shown above. (b) Selected outlines of the best preserved footprints, for each recognized morphotype, are reported.

*Figure 5 continued on next page*

*Figure 5 continued*

DOI: https://doi.org/10.7554/eLife.45204.008

The following figure supplement is available for figure 5:

**Figure supplement 1.** Loadings for the first three principal components.

DOI: https://doi.org/10.7554/eLife.45204.009

*Ursus* sp. hibernation areas are still recognizable with well-preserved nests of both cubs and adult bears.

## Preservation

Footprints are preserved in several areas of the cave, particularly in the innermost chamber ('*Sala dei Misteri*') and in the main gallery ('*Corridoio delle impronte*' – Footprints Corridor), which is divided into two corridors at different elevations of about 5 m (referred to as the lower and upper corridor, respectively) (*Figure 1*).

Flooding dynamics and cave geometry produced two different situations for sediment deposition and transport inside the cave. Detrital sediment comprising silty clay and well-sorted sandy sediment are most abundant on the floor of the '*Sala dei Misteri*'. Coarse lithologies comprising gravel-sized and larger (>2 mm) grains include a few fragments of bear bones. The sandy fraction comprises allogenic, surface-derived siliciclastic sediment. The '*Sala dei Misteri*' appears to have undergone episodic filling and erosion as a result of catastrophic storms.

Sediment in the '*Corridoio delle impronte*' comprises a large mud fraction and includes many coarse lithic fragments which are mainly carbonates (calcite and dolomite), suggesting an autogenic origin. Here, the trampled substrate is poorly consolidated and superimposed on a stalagmite crust. At the time when humans and other large mammals left their traces the cave substrate differed in different areas of the cave. In some areas, the substrate was plastic and in other areas it was water-logged or submerged. Differing moisture content of the substrate accounts for the variable preservation of detail of the tracks (e.g. registration of plantar arch, heel and metatarsal regions, digit tips, track walls), particularly of the associated extra-morphologies (e.g. expulsion rims, slipping traces). The surface of the substrate and the footprints are cross-cut by mud cracks, suggesting a loss of moisture in sediments after trampling. Carbonate crusts (comprising both calcite and dolomite) cover many of the footprints in areas subjected to more intense dripping. Iron and manganese oxide coatings were found in the crust, probably due to repeated immersion in ponded water.

## Results

Geometric morphometry performed on human footprints highlighted five main morphotypes (hereafter Morphs.) indicating a possible minimum number of five individuals entering the cave (*Figure 5*). This number is confirmed by the construction of morphological groups (*Table 2*) reconstructed through the overlapping of footprints that show a variability of less than 2% of the main parameters.

Morphs. 1 and 2 can be easily distinguished on the basis of the absolute footprint size. Morph. 1 includes footprints with a length of $13.55 \pm 0.49$ cm showing characters indicative of an early ontogenetic stage of the producer, such as digit traces and the heel area proportionally wider than those of the longer tracks. Morph. 2, with a length of 17 cm, is distinguished from Morph. one on the basis of a more pronounced plantar arch. Morph. 3 comprises footprints $20.83 \pm 0.51$ cm in length (*Figure 5* and *Table 3*). The plantar area is characterized by a very pronounced medial embayment. This corresponds with a strongly convex external margin, as is shared together with a strong adduction of digit I trace, an overall larger divarication of digit traces and a consistent separation between adjacent digits II-III and IV-V.

Morph. 4 (*Figures 5* and *6*) is represented by larger footprints ($22.80 \pm 0,42$ cm in overall length) with roughly straight medial and lateral margins and a medial embayment less marked than that of Morph. 3. Digit tip traces are strongly aligned and oriented forward. Morph. 5 (*Figures 5* and *7*) includes footprints of $25.73 \pm 0.45$ cm in overall length and slightly concave margins, with a variably pronounced plantar embayment. The footprints of Morph. 5 are generally more robust and

**Table 2.** Footprint shape features after 'Robbins footprint recording form' (1985, p.97–102).

| ID | Left (L) or right (R) foot | General appearance | | Relative length of toes | Toes region general appearance length - width | Toe one position | Ball region, general appearance length-width | Arch region | | Heel region, general appearance | Hell posterior margin | |
| --- | --- | --- | --- | --- | --- | --- | --- | --- | --- | --- | --- | --- |
| | | Length | Width | | | | | Medial margin | Lateral margin | | | |
| SM3 | R | short | broad | 1, 2, ? | short - broad | extended, anteriorly | | straight | concave | circular | convex pronounced | Morphotype 1 |
| SM4 | L | | | 1,2,3,4,5 | short - broad | extended, anteriorly | short - moderate | | concave | | | |
| SM43 | L | short | broad | 1,2, ? | short - broad | extended, anteriorly | | straight | convex | | convex pronounced | |
| SM17 | R | short | broad | 1,2,3,4,5 | short - broad | extended, anteriorly | | straight/concave | convex | | convex pronounced | |
| SM5 | R | moderate | moderate | 1,2,3,4,5 | short - broad | extended oblique medially | long - moderate | concave | concave | | convex slight | Morphotype 2 |
| SM42 | R | moderate | | 1,2, ? | | extended oblique medially | | concave | | | convex pronounced | |
| SM26 | L | moderate | broad | 1,2,3,4,5 | short - moderate | extended oblique laterally | | concave | concave | | convex slight | |
| CA8 | R | long | moderate | 1,2,3,4,5 | moderate - broad | extended oblique medially | long - moderate | concave | straight | oblong | convex pronounced | Morphotype 3 |
| CA10 | R | long | moderate | 1,2,3,4 | moderate - broad | extended oblique medially | | straight/concave | straight | oblong | convex pronounced | |
| SM15 | L | long | moderate | 2,3,1,4,5 | moderate - broad | extended oblique medially | long - narrow | concave | straight | oblong | convex pronounced | |
| SM11 | R | long | moderate | 1,2,3,4,5 | short - broad | extended anteriorly | | straight/concave | convex | oblong | convex pronounced | |
| SM6 | L | long | moderate | 2,1,3,4,5 | short - broad | flexed slight | long - narrow | concave | convex | oblong | convex pronounced | |
| SM1 | L | long | moderate | 2,3,1,4,5 | short - broad | extended oblique medially | moderate - narrow | unknown | convex | circular | convex pronounced | |
| C33 | L | long | broad | 1,2,3,4,5 | long - broad | extended oblique medially | long - broad | concave | convex | oblong | convex pronounced | |
| C36 | L | long | very narrow | 1,2,3,4,5 | moderate - narrow | extended oblique laterally | long - narrow | concave | straight | oblong | convex pronounced | |
| CA1 | R | long | | 1,2,3,4 | moderate - broad | extended anteriorly | moderate - narrow | straight | | oblong | convex, pronounced | Morphotype 4 |
| CA2 | L | long | moderate | 1,2,3,4,5 | moderate - broad | extended anteriorly | moderate - narrow | concave | straight | oblong | convex pronounced | |
| C61 | L | long | moderate | 1,2,3,4,5 | moderate - broad | extended oblique medially | moderate - narrow | straight/concave | straight/convex | oblong | convex pronounced | |
| C63 | R | long | moderate | 1,2,3,4,5 | moderate - broad | extended anteriorly | moderate - narrow | straight/concave | straight/concave | oblong | convex pronounced | |
| M21 | R | long | moderate | 1,2,3,4,5 | moderate - broad | extended anteriorly | moderate - narrow | concave | straight | oblong | convex pronounced | |
| C9 | R | long | moderate | 1 | | extended anteriorly | moderate - narrow | straight | convex | oblong | convex pronounced | |

*Table 2 continued on next page*

*Table 2 continued*

| ID | Left (L) or right (R) foot | General appearance | | Relative length of toes | Toes region general appearance length - width | Toe one position | Ball region, general appearance length-width | Arch region | | Heel region, general appearance | Hell posterior margin | |
|---|---|---|---|---|---|---|---|---|---|---|---|---|
| | | Length | Width | | | | | Medial margin | Lateral margin | | | |
| C44b | L | long | broad | 2,1,3,4,5 | short - broad | extended oblique medially | moderate - broad | concave | convex | oblong | convex, moderate | |
| C60 | L | long | broad | 1,2,3,4,5 | moderate - broad | extended oblique laterally | moderate - broad | straight/ concave | straight/ concave | circular | convex moderate | Morphotype 5 |
| C37 | L | long | broad | 1,2,3,4,5 | moderate - broad | extended anteriorly | moderate - broad | straight/ concave | straight/ concave | circular | convex moderate | |
| C35b | R | long | broad | 1,2,3,4,5 | moderate - broad | extended anteriorly | moderate - broad | | straight/ concave | circular | convex moderate | |
| C44 | L | long | broad | 1,2,3,4,5 | moderate - broad | extended anteriorly | moderate - broad | concave | straight/ concave | oblong | convex pronunced | |

DOI: https://doi.org/10.7554/eLife.45204.010

stockier than those of Morphs. 3 and 4, sharing with Morph. 4 the straight, forwardly oriented digit tips, and with Morph. 3 an adducted digit I trace.

Plantigrade tracks enabled us to estimate stature, weight and ontogenetic stage of the producers based on biometric measurements (*Table 3*) and the adopted formulas (see Methods). An estimation of the gender for Morph. 5 was also attempted (see Methods).

The group of human track producers entering the cave comprised: a three-year-old child about 88 cm tall (Morph. 1); a child at least 6 years old and about 110 cm tall (Morph. 2); a pre-adolescent, between 8 and 11 years old, about 135 cm tall (Morph. 3); a sub-adult to adult about 148 cm tall (Morph. 4); and an adult about 167 cm tall (Morph. 5). Estimate of the stature for the Morph. 5 is further supported by the length of the tibia derived from the available kneeling traces (see Methods). Our results concerning Morphs. 4 and 5, which are referred to adult individuals, are in agreement with the average stature of European Upper Palaeolithic people (162.4 ± 4.6 cm for males and 153.9 ± 4.3 cm for females) (*Villotte et al., 2017*).

Body mass estimates derived from footprint parameters suggests slender and muscular body proportions for all the trackmakers. Arch angle and footprint morphology suggests a male as the probable trackmaker of the largest footprint group. Differently, the gender result difficult to infer for morphotypes 1, 2, 3, 4, although Morph. 4 can be referred most probably to a female.

Digitigrade and semi-plantigrade footprints provide information on pedal postures and the behavior of the producers passing through different sub-environments of the cave. Both these footprint types were, in most cases, traced back to the same type of producer by comparison with complete footprints indicating complete foot support during locomotion. Some semi-plantigrade footprints (e.g. *Figure 8*, C44, C44b) show a strongly adducted trace of digit I and an apparent alignment with the other digits, probably because of the intra-rotation movements of the distal portion of the foot during the thrust phase. Footprints included in the Morph. 3 show a peculiar pedal morphology. While the resulting morphology of the digit I trace is explained by walking on a waterlogged substrate, the separation between digit pairs II-III and IV-V suggest an inherited familiar trait or a pathological condition of the producer's feet. The producer was not incapacitated, showing the greatest mobility in the hypogeal environment.

In the lower corridor (*Figure 9*), a few of these footprints are associated with elongated traces imprinted by the producers' knees resting on the substrate. Successions of kneeling traces can be clearly recognized for Morphs. 3, 4, and 5. Based on the overall size of the metatarsal and knee couples aligned on the substrate, crawling in a totally unknown environment is inferred for the whole group. These knee imprints (e.g. *Figure 8*, C42) show the muscle structure of the knee joint and adjacent regions. The patella, the patellar ligament (tendon), the tibial tuberosity, the fibular head,

**Table 3.** Measurements and elaboration data (foot index, stature, body mass and age) based on the best-preserved tracks from the 'Sala dei Misteri' and 'Corridoio delle impronte'.

*Body mass: (a) *Citton et al., 2017*; (b) *Bavdekar et al., 2006*; (c) *Grivas et al., 2008* (see text).

| ID | Left (L) or right (R) foot | LENGTHS FOOT Dt1 (cm) | Dt2 (cm) | Dt3 (cm) | Dt4 (cm) | Dt5 (cm) | BALL medial (mtm-BL) (cm) | BALL lateral (mtl-BL) (cm) | ARCH medial (ntu-BL) (cm) | ARCH lateral (mttu-BL) (cm) | HEEL medial (ccm-BL) (cm) | HEEL lateral (ctul-BL) (cm) | WIDTHS BALL mtm-horiz (cm) | ARCH mttu-horiz (cm) | HEEL ctul-horiz (cm) | ANGLES T1-T5 (degrees) | Angle of toe declination | max FL (cm) | max FW (mtm-mtl) (cm) | arc angle (degrees) | Foot index | stature (cm) | body mass* (kg) | age | Morphotype |
|---|---|---|---|---|---|---|---|---|---|---|---|---|---|---|---|---|---|---|---|---|---|---|---|---|---|
| SM3 | R | 13 | 12.3 | | | | 10.5 | 9.2 | 5.5 | 6.8 | 2.3 | 2.4 | 5.5 | 4.7 | 4.3 | | | 13 | 6 | 20 | 0.46 | 84.36 | 11.78[a] | | Morpho type 1 |
| SM4 | L | 13.5 | 12.5 | 11.5 | 11 | 9,8 | 10.2 | 8 | 4 | 5.5 | 2.2 | 1.6 | 5.5 | 4 | 4.2 | 40 | | 13.5 | 6.5 | 22 | 0.48 | 87.61 | 12.55[a] | | |
| SM43 | L | 13.5 | 13.5 | | | | 10.5 | 9.4 | 5.5 | 6.5 | 2.2 | 1.8 | 6 | 5.5 | 4.6 | | | 13.5 | 6.5 | 20 | 0.48 | 87.61 | 12.55[a] | | |
| SM17 | R | 14.2 | 13.8 | 13.5 | 12.6 | 10.8 | 10.2 | 8.5 | 5 | 6 | 2 | 1.8 | 6.5 | 4.8 | 4 | 32 | | 14.2 | 6.8 | 25 | 0.48 | 92.15 | 13.70[a] | | |
| | | | | | | | | | | | | | | | | | | 13.55 ± 0.49 | | | 0.48 ± 0.01 | 87.93 ± 3,20 | 12.64 ± 0,79[a] | <3 | |
| SM5 | R | 17 | 16.8 | 15.5 | 14 | 13 | 12 | 12 | 5.5 | | 2,5 | 2 | 6.5 | | 5 | 35 | | 17 | 6.8 | 28 | 0.40 | 110.32 | 19.50[a] | | Morpho type 2 |
| SM42 | R | 17 | 16.8 | | | | 12.5 | 11.5 | 6.5 | 10 | 3.4 | 2.5 | 7 | | 5.4 | | | 17 | 7.2 | 25 | 0.42 | 110.32 | 19.50[a] | | |
| SM26 | R | 18 | 16.5 | 15 | 14 | | | | | | 2 | 3 | | | 5 | | | 18 | | 28 | - | 116.81 | 22.12[a] | | |
| | | | | | | | | | | | | | | | | | | 17 | | | 0.41 ± 0,02 | 110.32 | 19.5[a] | 5–6 | |
| CA8 | R | 20.2 | 19.8 | 19.4 | 18.3 | 16.8 | 15 | 13 | 7.5 | 9.2 | 2.6 | 2 | 7.5 | 6 | 4.5 | 30 | | 20.2 | 8 | - | 0.40 | 131.08 | 29.18[a] | 8–10 | Morpho type 3 |
| C10 | R | 20.5 | | 19.2 | 18.4 | 16.5 | 15.5 | 12.5 | 6.5 | 8.5 | 2.8 | 2.6 | 7 | 5.5 | 5.6 | 32 | | 20.5 | 8 | - | 0.39 | 133.03 | 30.30[a] | boy/9–11 girl | |
| SM15 | L | 20.5 | 20.5 | 18.5 | 17.5 | 16 | 15.4 | 13 | 6.5 | 7.8 | 4 | 2 | 7 | 4 | 6 | 30 | | 20.5 | 7 | 45 | 0.34 | 133.03 | 30.30[a] | | |
| SM11 | R | 21 | 20.2 | 19.7 | 18 | 16.8 | 15.4 | 14.5 | 5.5 | 8 | 3.3 | 2.3 | 7.5 | 6.8 | 6 | 30 | | 21 | 7.5 | 40 | 0.36 | 136.28 | 32.28[a] | | |
| SM6 | L | 21 | 21.5 | 20.5 | 18.8 | 17 | 17.5 | 15.5 | 6.8 | 8.5 | 4 | 2.2 | 9.2 | 5.5 | 5.7 | 20 | | 21.5 | 9 | 40 | 0.42 | 139.52 | 34.38[a] | | |
| SM1 | L | 21.2 | 21.3 | 20.8 | 20.4 | 18.5 | 17.4 | 14.8 | 6.8 | 8.7 | 3.8 | 2.5 | 8 | 5 | 6 | 20 | | 21.3 | 8.5 | 40 | 0.40 | 138.22 | 33.52[a] | | |
| C33 | L | 22.2 | 21 | 19.5 | 17.3 | 14.8 | 15.5 | 10.7 | 6 | 7.8 | 2.5 | 2 | 10 | 6.5 | 5.8 | 45 | | 22.2 | 10.5 | 48 | 0.47 | 144.06 | 37.55[a] | | |
| C36 | L | 22.7 | 21.7 | 19.5 | 17.8 | 16.2 | 14.5 | 14.5 | | | 3 | 2.5 | | | 5.5 | 48 | | 22.7 | | - | - | 147.31 | 39.99[a] | | |
| | | | | | | | | | | | | | | | | | | 20.83 ± 0.51 | | | 0.38 ± 0.03 | 135.19 ± 3.33 | 31.66± 2.05[a] | 8–11 | |

*Table 3 continued on next page*

*Table 3 continued*

| ID | Left (L) or right (R) foot | LENGTHS — FOOT Dt1 (cm) | Dt2 (cm) | Dt3 (cm) | Dt4 (cm) | Dt5 (cm) | BALL medial (mtm-BL) (cm) | BALL lateral (mtl-BL) (cm) | ARCH medial (ntu-BL) (cm) | ARCH lateral (mttu-BL) (cm) | HEEL medial (ccm-BL) (cm) | HEEL lateral (ctul-BL) (cm) | WIDTHS — BALL mtm-horiz (cm) | ARCH mttu-horiz (cm) | HEEL ctul-horiz (cm) | ANGLES — Angle of toe declination T1-T5 (degrees) | max FL (cm) | max FW (mtm-mtl) (cm) | arc angle (degrees) | Foot index | stature (cm) | body mass* (kg) | age | Morphotype |
|---|---|---|---|---|---|---|---|---|---|---|---|---|---|---|---|---|---|---|---|---|---|---|---|---|
| CA1 | R | 22.4 | 21.8 | 20.8 | 18.5 | | 14.8 | 12.8 | 7 | | 3.1 | | 8 | | | 30 | 22.4 | 8.5 | - | 0.38 | 145.36 | 45.48[b] - 46.66[c] | | Morphotype 4 |
| CA2 | L | 22.5 | 22 | 21 | 20 | 18.5 | 17.3 | 15.2 | 6 | 8.6 | 3.5 | 3.5 | 8.8 | 6 | 6.4 | 30 | 22.5 | 8.5 | 44 | 0.38 | 146.01 | 45.66[b] - 47.19[c] | | |
| C61 | L | 23 | 21.7 | 20.8 | 20 | 19.5 | 16.4 | 14 | 8.5 | 9.5 | 3.8 | 3 | 7.5 | 6.5 | 5.4 | 30 | 23 | 9 | 45 | 0.39 | 149.25 | 46.57[b] - 49.82[c] | | |
| C63 | R | 23.3 | 22.2 | 20.8 | 19.8 | 18.7 | 17 | 14 | 7.9 | 9.5 | 3.5 | 3.8 | 8.5 | 6.7 | 5.3 | 32 | 23.3 | 9 | 42 | 0.39 | 151.20 | 47.12[b] - 51.39[c] | | |
| M21 | R | - | 21.6 | 21.3 | 20.7 | 19.2 | 16.7 | 14.6 | 7.8 | 10.5 | 3.5 | 2.4 | 9.6 | 4.8 | 6.2 | | - | 9.8 | 42 | - | - | - | | |
| C9 | R | 22.5 | | | | | | | | | | | 8 | | 5.5 | | 22.5 | 8 | - | 0.36 | 146.01 | 45.66[b] - 47.19[c] | | |
| C44b | L | 21.5 | 21 | 20 | 19.5 | 18.5 | 15 | 13 | 7 | 4 | 2.2 | 1.5 | 10 | 5 | 5.5 | 20 | 21.5 | 10.5 | 50 | 0.49 | 139.52 | 43.84[b] - 41.93[c] | | |
| | | | | | | | | | | | | | | | | | 22.80 ± 0.42 | | | 0.38 ± 0.01 | 147.96 ± 2.75 | 46.21 ± 0.77[b] - 48.76 ± 2.23[c] | >14 - adult | |
| C60 | L | 25.3 | 24.2 | 22.7 | 21.4 | 20 | 18 | 14.8 | 6 | 8 | 3.5 | 3.3 | 10.5 | 7.5 | 6.4 | 35 | 25.3 | 11 | 52 | 0.43 | 164.18 | 50.76[b] | | Morphotype 5 |
| C37 | L | 25.7 | 23.8 | 22.5 | 21 | 19.5 | 18.4 | 14.7 | 6 | 7.8 | 3.7 | 3.5 | 10.5 | 7.5 | 7 | 35 | 25.7 | 10.5 | 55 | 0.41 | 166.77 | 51.48[b] | | |
| C35b | R | 26.2 | 24.8 | 22.8 | 20.7 | 18.7 | 17 | 13.7 | 5.8 | 7.2 | 3.7 | 2.7 | 9.5 | 7 | 6.7 | 40 | 26.2 | 10.5 | - | 0.40 | 170.02 | 52.39[b] | | |
| C44 | L | 25 | 23.5 | 22.5 | 20.8 | 19 | 16.8 | 14.2 | 5.5 | 6.7 | 2.8 | 2 | 9.2 | 5.7 | 6 | 45 | 25 | 10 | 50 | 0.40 | 162.23 | 50.21[b] | | |
| | | | | | | | | | | | | | | | | | 25.73 ± 0.45 | | | 0.41 ± 0.02 | 166.99 ± 2.93 | 51.54 ± 0.82[b] | >14 - adult | |

DOI: https://doi.org/10.7554/eLife.45204.011

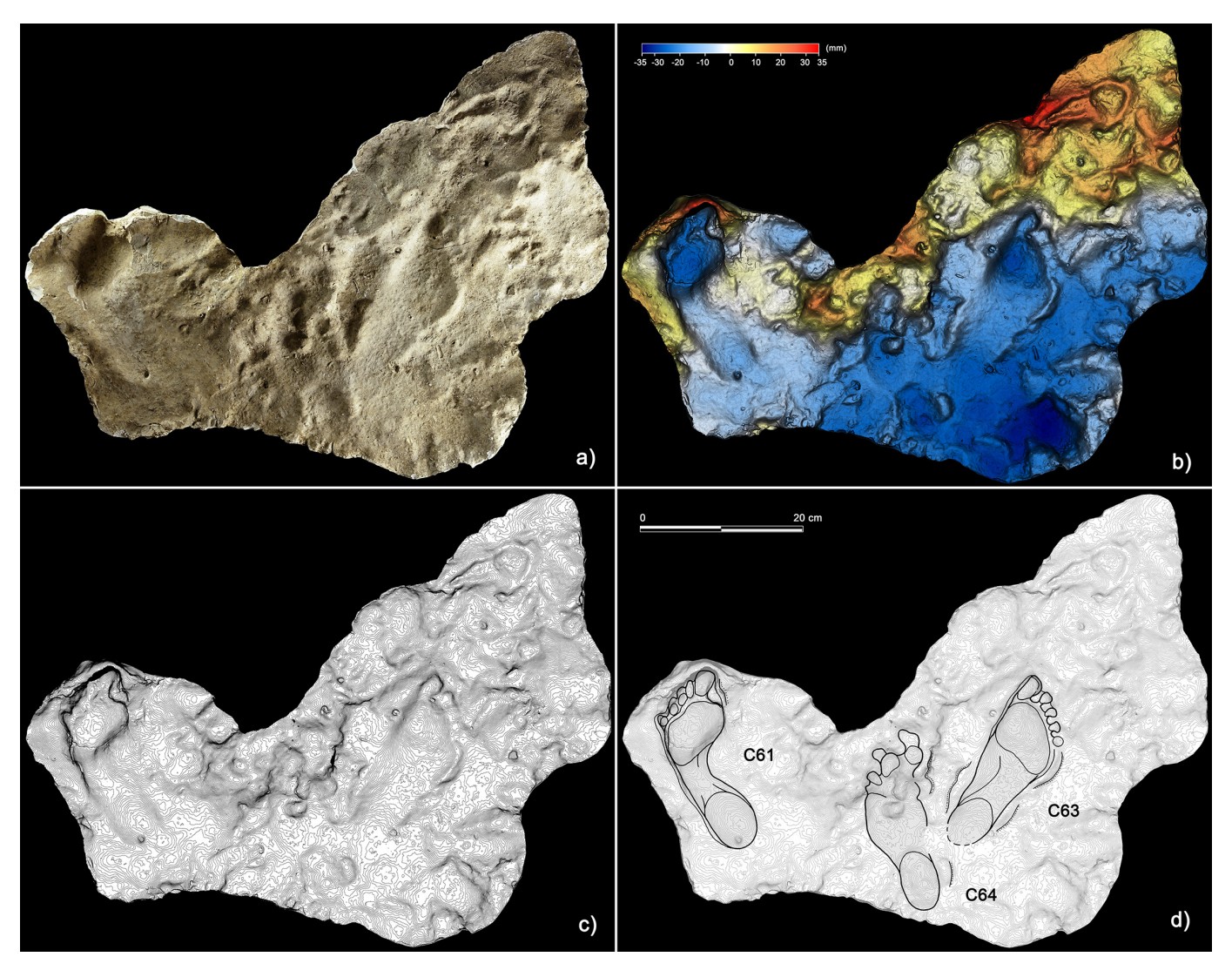

**Figure 6.** Plantigrade tracks from the 'lower corridor'. (**a**) Cast of the 1950s reproducing tracks C61, C63 and C64, preserved in the sector A of the 'lower corridor' (see *Figure 1* main text). (**b**) Digital terrain model of the cast obtained from the HDI 3D Scanner. (**c**) Topographic profile with contour lines, obtained from b. (**d**) Interpretive draw. Note that the tracks C61 and C63 were most likely left by a producer (Morph. 4) crouched against the side-wall of the 'lower corridor'.

DOI: https://doi.org/10.7554/eLife.45204.012

the basis of the vastus medial and the iliotibial band are recognizable and allow us to infer the body structure of the trackmakers.

## Discussion

Successions of kneeling traces allowing one to infer a crawling locomotion for the trackmakers have never been reported before. Isolated kneeling traces (i.e. a footprint followed by a knee imprint of the same leg) have only previously been reported from the 'Galerie Wahl' at Fontanet cave and from the 'Salle des Talons' at Tuc d'Audoubert cave, France (*Pastoors et al., 2015*) but are not sufficient to infer crawling. In addition, the anatomic details clearly recognizable in the crawling traces from 'Grotta della Bàsura' enabled us to hypothesize that no clothing was interposed between the limb and the trampled sediments.

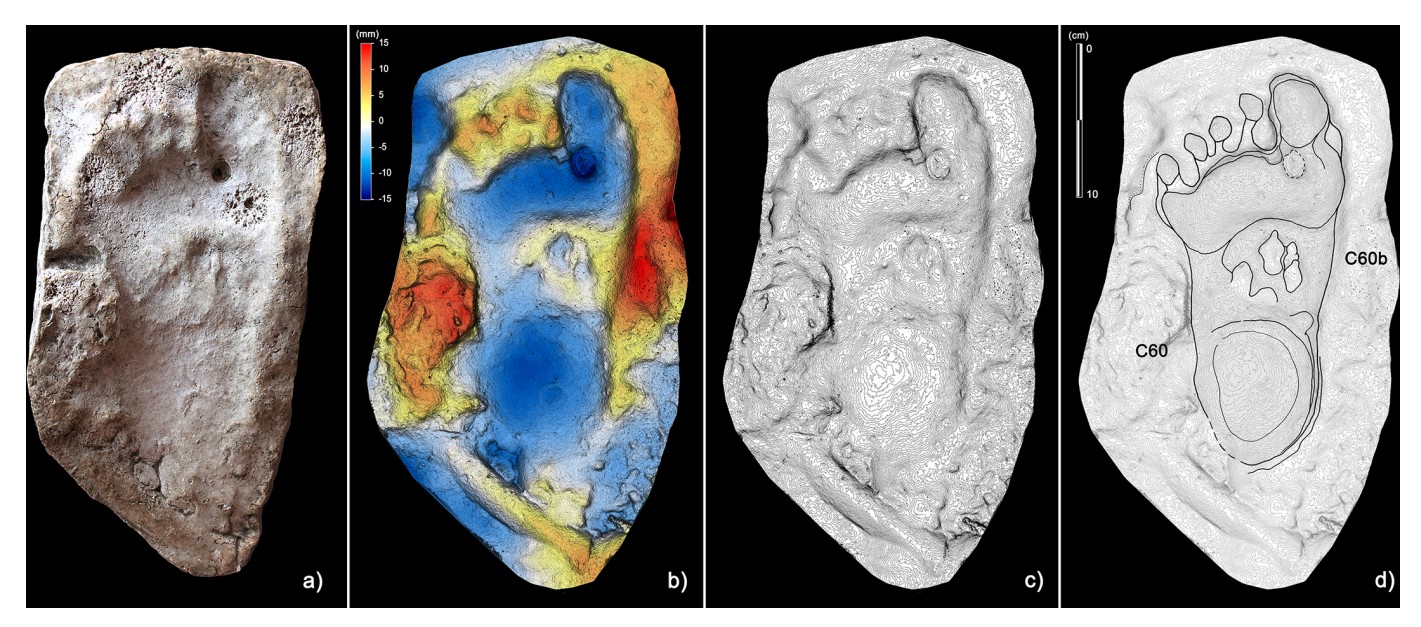

**Figure 7.** Plantigrade track from the 'lower corridor'. (a) Cast of the 1950s reproducing the track C60 preserved in the sector A of the 'lower corridor' (see *Figure 1* main text). (b) Digital terrain model of the cast obtained from the HDI 3D Scanner. (c) Topographic profile with contour lines, obtained from b. (d) Interpretive draw. A superimposed partial canid track, C60b, is clearly recognizable in the metatarsal area of the human footprint (Morph. 5).
DOI: https://doi.org/10.7554/eLife.45204.013

The integration of all available ichnological evidence with data on the complex morphology of the cave enabled us to reconstruct a detailed hypothesis for the events which took place by Paleolithic peoples while exploring the cave about 14,000 BP.

The short radiometric interval, documented by both previous radiocarbon dates and those derived from the latest research presented here (*Table 1*), together with interferences relating to the interrelations of the different footprints, suggest that all individuals entered the cave at the same time. In particular, the group of footprints C34, C35b, C36, C35 in the terminal part of the *'Corridoio delle impronte'* provides relative timing of the impressions: C35 (Morph. 4) is superimposed on C36 (Morph. 3) and on C34 and C35b (both Morph. 5) (*Figure 9*, d3). In this specific case, the producer of Morph. 4 passed after the producers of Morphs 3 and 5, respectively. In the entrance of the same corridor, the timing is reversed: footprint C63 (Morph. 4) is superimposed by C64 (Morph. 3) proving that Morphs. 3 and 4 were made at the same time (*Figure 10*).

It is also difficult to assume that Morphs. 1 and 2 may have entered the deepest part of the cave without the presence of some older individual: if the older individuals who were leading the exploration were either Morphs. 3 or 4 the group consequently increases to include at least four individuals. No relationship can be established for the larger individual; however, as reported below, the fact that the footprints of this individual are regular and were closely followed on the same path by all the other individuals, suggests that the larger one was likely the leader of the synchronous exploration. Thus, all the available lines of evidence, particularly the complex interrelationship of the studied footprints, strongly suggest that a single exploration event by a heterogeneous group of five individuals is the most parsimonious and best supported hypothesis.

Five individuals, comprising two adults, an adolescent and two children, entered the cave barefoot and with a set of *Pinus* t. *sylvestris/mugo* bundles which were burned for illumination purposes. The adopted lighting system of several sticks enabled a longer period of lighting, as inferred from the fire wood illumination bundles adopted by the Bronze age salt miners at Hallstatt (*Grabner et al., 2007*; *Grabner et al., 2010*). Lighting bundles were usually made of resinous wood (Scots pine or Mountain pine), and called torch wood (*Ast, 2001*; *Théry-Parisot et al., 2018*). This interpretation fits with the archaeological evidence from the Bàsura Cave.

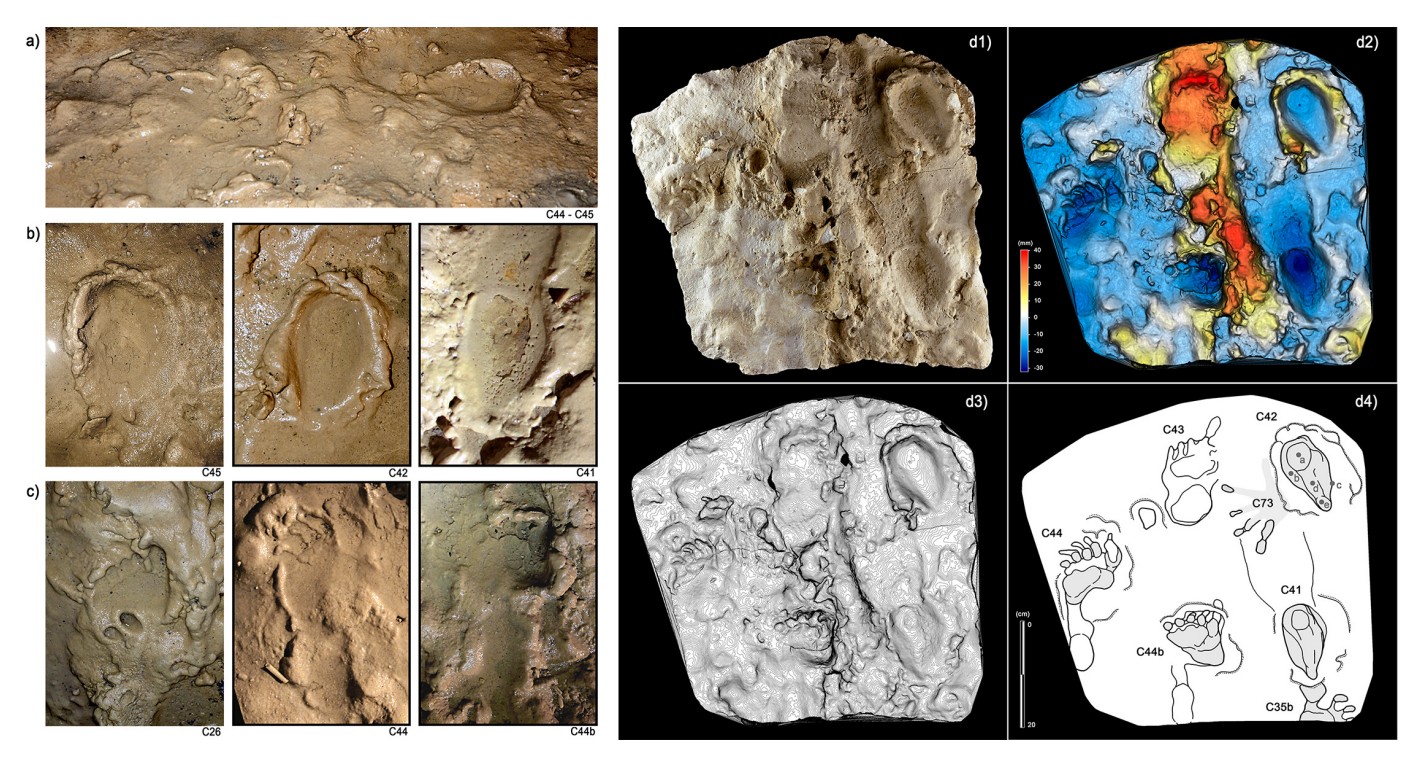

**Figure 8.** Selection of semi-plantigrade and knee traces from the 'lower corridor' of the '*Corridoio delle impronte*' in the Bàsura cave, indicating crawling locomotion of the producers. (a) Associated metatarsal (C44) and knee (C45) traces allowing estimation of the tibial length of the producer. (b) Knee traces (C45, C42 and C41) imprinted on a plastic, waterlogged muddy substrate. (c) Metatarsal traces (C26, C44 and C44b) imprinted on a plastic, waterlogged muddy substrate. (d1) cast of the 1950s reproducing two knee (C41 and C42) and two metatarsal (C44, C44b) traces preserved in the area B of the 'lower corridor' (see *Figure 1*). (d2) Digital Terrain Model obtained from the HDI 3D Scanner. (d3) Topographic profile with contour lines, obtained from d2. (d4, Interpretive draw. In the knee trace, C42 are located the impressions of the patella (a), vastus medials (b), the fibular head (c), the patellar ligament (d) and the tibial tuberosity (e).

DOI: https://doi.org/10.7554/eLife.45204.014

After a walk of approximately 150 m from the original opening of the cave and a climb of about 12 m, the group arrived at the '*Corridoio delle impronte*'. They proceeded roughly in single file, with the smallest individual behind, and walked very close to the side wall of the cave, a safer approach also used by other animals (e.g. Canidae *incertae sedis* and bears) when moving in a poorly lit and unknown environment. The slope of the tunnel floor, inclined by about 24°, may have further forced the individuals to proceed along the only flat area in the lower corridor, a couple of meters from the left wall of the cave. About 10 m from the '*Corridoio delle impronte*', the cave roof drops to below 80 cm and members of the group were forced to crawl (*Figure 11B*), placing their hands (*Figure 12*) and knees (*Figure 8b*) on the clay substrate (*Figure 9*) (see also *Video 1*).

After a few meters, the group leader stopped, impressing two parallel calcigrade footprints, possibly to decide on the next movement and proceeded to cross the parts where the cave roof was at its lowest. The other individuals also stopped at the same place as the leader, and then proceeded along the same path by crawling and following the group leader, as indicated by the timing reconstructed from interactions between the tracks (*Figure 9*, d3).

After passing a bottleneck of blocks and stalagmites, the party descended for about ten meters along a steeply sloping surface. The whole group traversed a small pond, leaving deep tracks on the plastic waterlogged substrate, climbed a slope of 10 m beyond the '*Cimitero degli orsi*', and finally arrived at the terminal room '*Sala dei Misteri*', where they stopped. On the walls, several charcoal traces, generated by the torches, are preserved.

Some charcoal handprints produced by a flexed reaching up more than 170 cm on the roof of the '*Sala dei Misteri*' confirm that the tallest individuals (Morphs. 4 and 5) were able to touch this part of

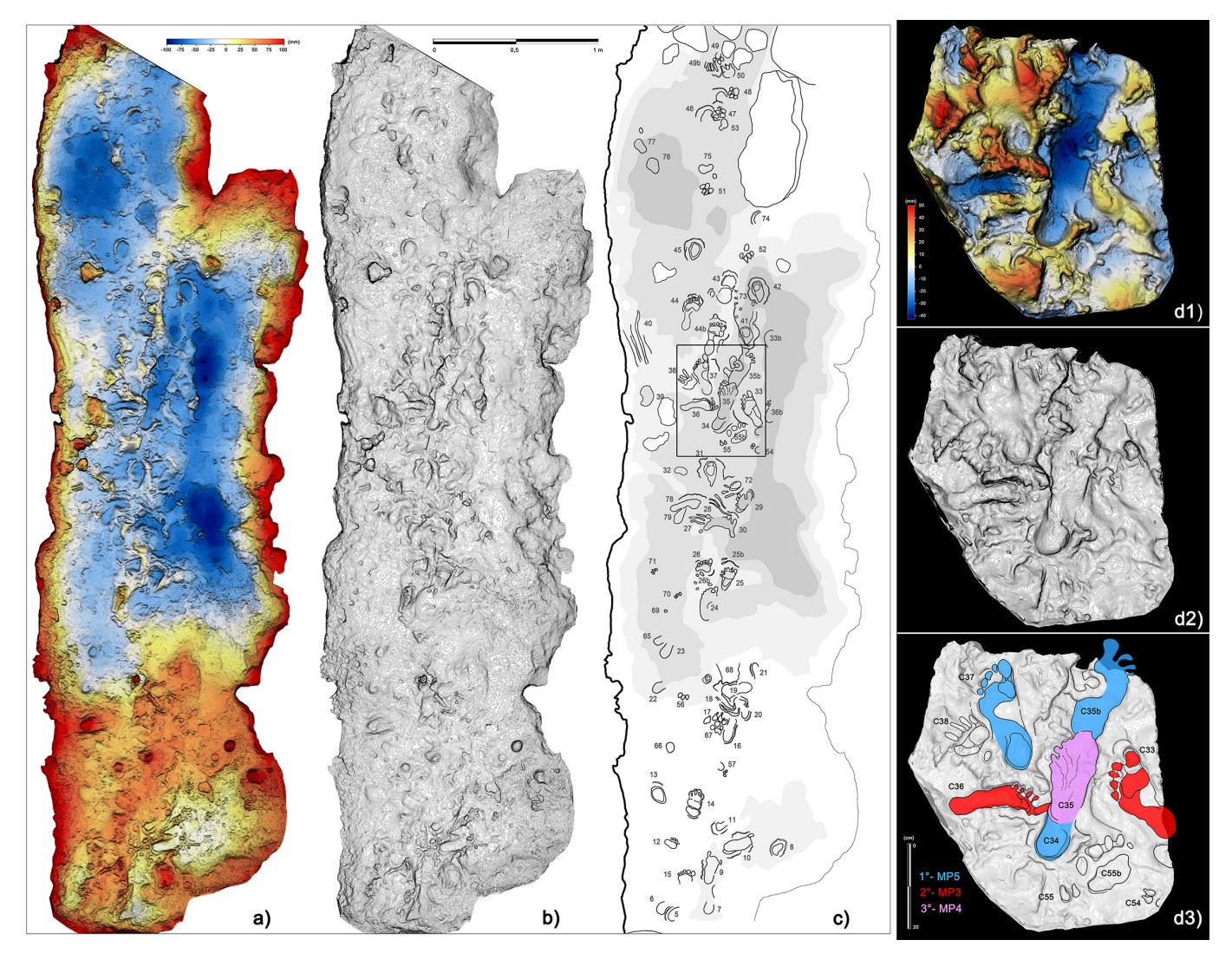

**Figure 9.** Crawling locomotion in the 'lower corridor' (sector B in *Figure 1*). (a) Color topographic profile obtained from the digital photogrammetric model. (b) Topographic contoured profile. (c) Interpretive draw of the track-bearing surface (numbers identify single tracks and traces and are to be intended as preceded by the letter C). (d1) Digital Terrain Model obtained from a cast of the 1950s reproducing a small area of the 'lower corridor'. (d2) Topographic profile with contour lines, obtained from d1. (d3) Interpretive draw and timing of the different recognized tracks.

DOI: https://doi.org/10.7554/eLife.45204.015

the gallery. The fact that their footprints are not preserved relates to the loss of the central portion of the hall floor. In the same room, the adolescent and children started collecting clay from the floor and smeared it on a stalagmite at different levels according to height, as suggested by the breadth and relative distribution of the finger flutings on the karst structure. During their sojourn in the innermost room of the cave the young individual, that produced Morph. 2, imprinted ten clear heel traces (*Citton et al., 2017*), which are here interpreted as calcigrade tracks produced by a trackmaker who is momentarily standing-still to excavate and manipulate clay as was also recorded for the 'Salle des Talons' at Tuc d'Audoubert cave (*Pastoors et al., 2015*).

After stopping for several minutes (considering the quantity and ubiquity of the tracks), they exited and followed a route which did not always adhere to that followed on entry. After passing the small pond, they crossed the upper corridor following a more comfortable and safer route (*Figure 11C*). It is important to note that in the upper corridor all the prints point in the direction of

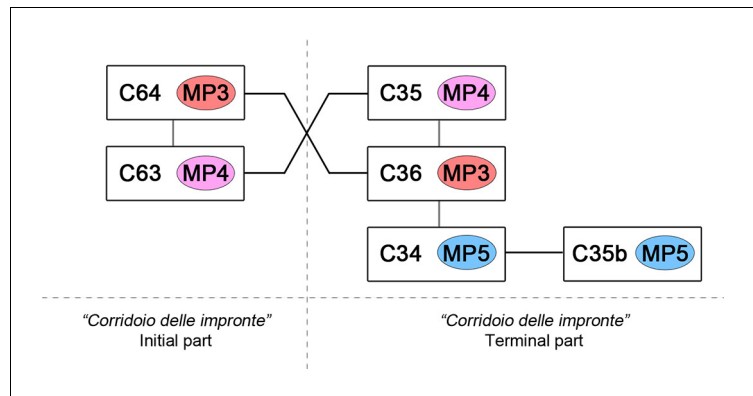

**Figure 10.** Timing of impressions of human footprints. The interference between footprints attributed to different individuals suggests a single exploring event of the cave. In particular the cross-overlapping of MP3 and MP4 trackmakers confirms their contemporary entry into the main gallery.
DOI: https://doi.org/10.7554/eLife.45204.016

the exit (*Figures 11C* and *13*) while in the lower corridor, with the axis of the foot oriented parallel to the walls, most of the footprints are directed toward the interior of the cave.

## Concluding remarks

A holistic analysis comprising several lines of inter-related ichnological evidence enabled reconstruction of several snapshots depicting a small and heterogeneous group of Upper Palaeolithic people that explored a cave about 14,000 years ago (*Video 1*). They traversed the uneven topography of the cave floor and carried out social activities in the most remote room, leaving evidence in their traces of a unique testimony to human curiosity.

The lower corridor was traversed on entry into the cave and documents the first unequivocal evidence of crawling locomotion in the human ichnological record. This mode of locomotion was adopted by the explorers to obviate variation in the height of the cave roof. As no outgoing footprints were documented in the lower corridor it appears that the group chose to exit through the upper corridor, as the cave roof is higher and the substrate firmer. An additional reason for choosing this path of exit could have been the exploratory and curiosity factor to follow a different and unexplored path to reach the cave exit.

Anatomical features clearly registered on the substrate indicate that the lower limbs of the individuals were not covered with clothing. Our study also confirms that very young children actively participated in the activities of the Upper Palaeolithic populations, even in seemingly dangerous tasks, such as the deep exploration of the cave environment lit only with torches. As recently suggested for other European caves (*Pastoors et al., 2015*; *Pastoors et al., 2017*; *Ledoux et al., 2017*) the 'Grotta della Bàsura' site strongly supports the hypothesis that the cave exploration in Upper Paleolithic was carried out by groups of heterogeneous age and gender.

The Epigravettian necropolis of the Arene Candide Cave (AMS dates spanning 12,820–12,420 cal BP for the first phase and 12,030–11,180 cal BP for the second phase), consisted of a 'mixed' sample of people (males, females, adults, children) and suggests an Upper Palaeolithic people composition very similar to that highlighted in the 'Grotta della Bàsura' (*Riel-Salvatore et al., 2018*; *Sparacello et al., 2018*). The burial of a newborn infant, together with grave goods, recently discovered in the Arma di Veirana cave (Erli, Savona, Liguria) situated in a valley 10 km from the coast, further indicates that women and children systematically followed the movements of the group in the territory (F. N. pers. obs.) (*Negrino et al., 2017*) and shared, at least in part, the activities of men and had similar personal adornment. The tracks left in the Bàsura indicate that the behavior of hunter-gatherers was not always driven by subsistence requirements, but as manifested by many ethnographic examples, also by fun and frivolous activities.

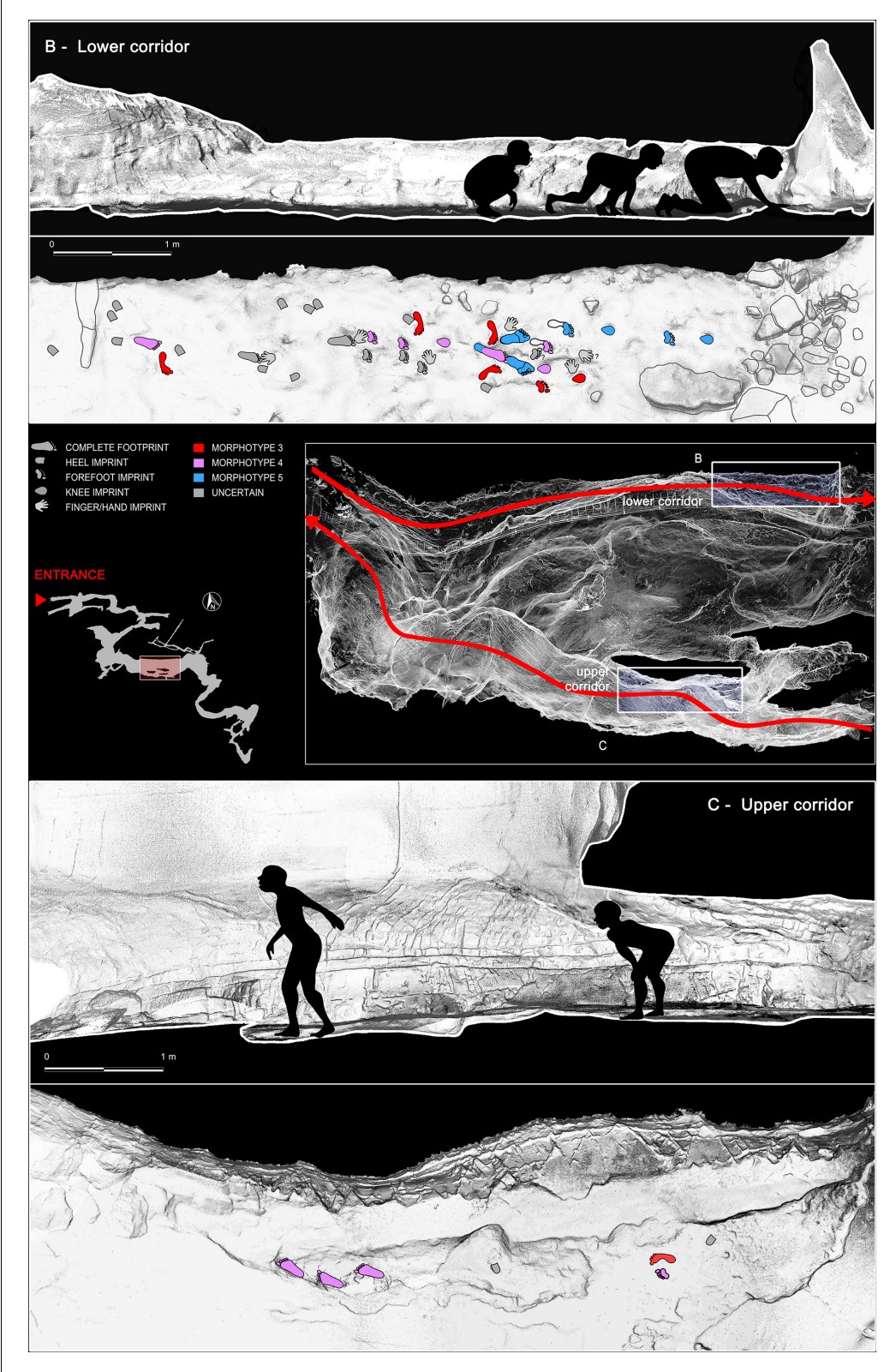

**Figure 11.** Reconstruction of the exploration routes chosen by the producers to enter and exit the cave. (B) Crawling locomotion adopted by the producers to cross the 'lower corridor' and access to the innermost rooms of the cave. (C) Exit route passing through the 'upper corridor', traveled by the producers in complete erect walking. The smallest producers are not reported in the sketch.
DOI: https://doi.org/10.7554/eLife.45204.017

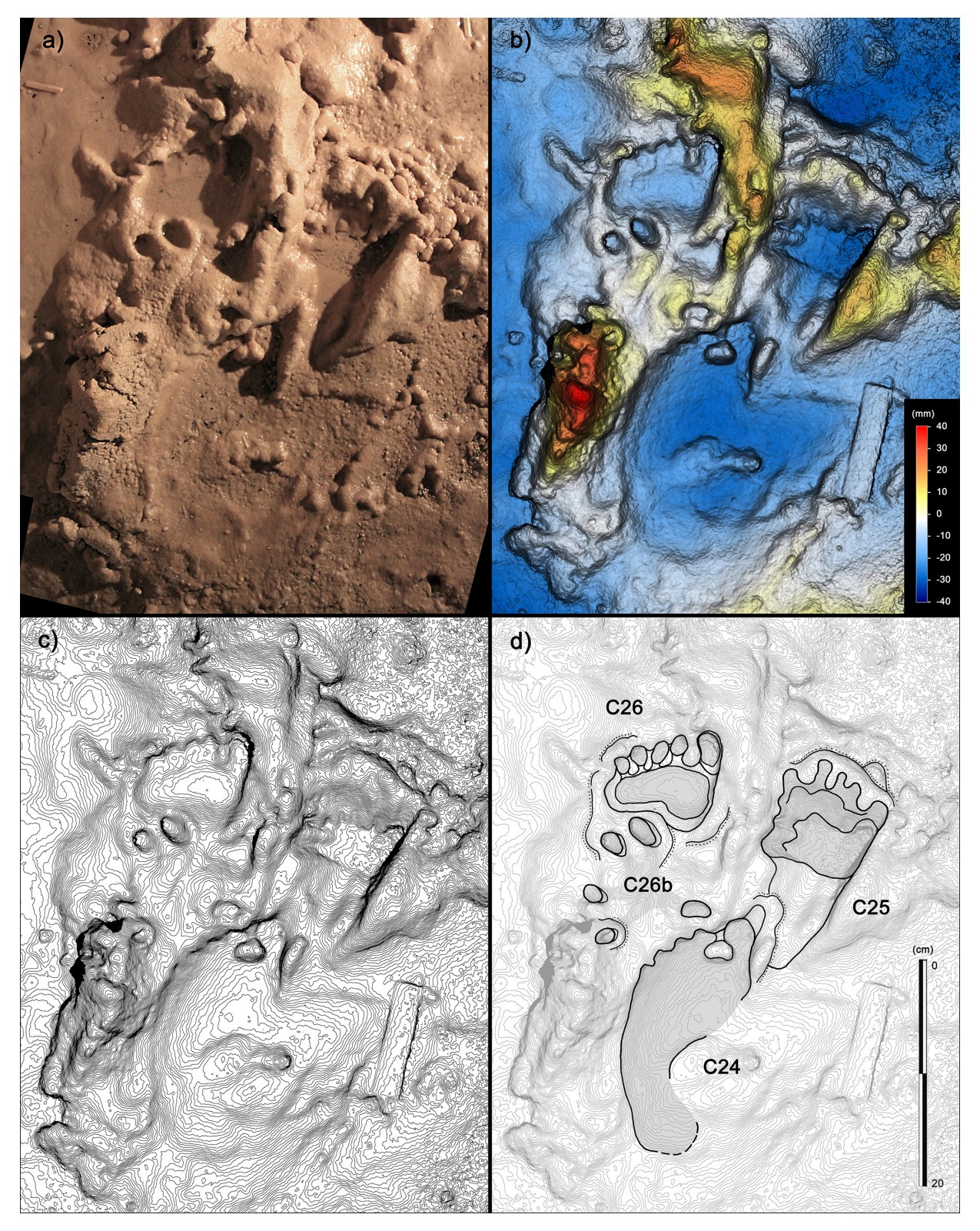

**Figure 12.** Human tracks from the 'lower corridor'. (**a**) Tracks C26, C26b, C25 and C24 from the sector B of the 'lower corridor' (see **Figure 1** main text). (**b**) Digital terrain model obtained from high-resolution photogrammetry. (**c**) Topographic profile with contour lines, obtained from b. (**d**) Interpretive draw. C26b is interpreted as a partial hand-print of which only digit traces are preserved, interfering with a metatarsal trace deeply imprinted on a muddy, highly plastic, substrate.

*Figure 12 continued on next page*

*Figure 12 continued*

DOI: https://doi.org/10.7554/eLife.45204.018

## Materials and methods

### Chronology

Radiometric dating of charcoals previously established the presence of humans in the cave to the Upper Palaeolithic, around 12,340 ± 160 years BP (*De Lumley and Giacobini, 1985*). The stalagmite crust preserving the footprints and incorporating fragments of coal is dated to between 14,300 ± 800 and 13,100 ± 500 (*Yokoyama et al., 1985*). The final phase of stalagmite growth, which closed the entrance and sealed the 'time capsule', occurred at 12,000 ± 1100 (*Yokoyama et al., 1985*).

New radiometric dating of charcoal samples of *Pinus* t. *sylvestris/mugo* was undertaken in 2017 at a AMS facility at Groningen (NL) on (*Table 1*). Material was collected from the trampled palaeo-surface during recent excavations inside the '*Sala dei Misteri*' (*Figure 14*).

### Laser scanning acquisition

Documentation of the sequence of events was contextualized and visualized on the rough cave topography through three-dimensional mapping of the cave performed by laser-scanning. The main landmarks of the cave were digitally recorded using the laser scanner ScanStation2 Leica and ScanStation C10 Leica. The scans were performed at 360° (acquisition grid of the point cloud of 2 × 2 cm a probe 7 m and in correspondence to the areas with the highest concentration of traces, an acquisition grid of 0.5 × 0.5 cm probe 7 m). In total, 23 stations were run (nine in the '*Sala dei Misteri*' and 14 in the '*Corridoio delle impronte*' areas); 38 targets (16 in the '*Sala dei misteri*' and 22 in the '*Corridoio delle impronte*') were used for the point clouds registration. The Leica Geosystems HDS Cyclone 9.1 software was used to process the data. The recording shows a final alignment error of 2 mm for the model of the '*Sala dei Misteri*' and 1 mm for the '*Corridoio delle impronte*' (*Video 1*). From the models, reliefs were obtained at various degrees of detail that enabled georeferencing of all traces. The original cast performed in 1950 were digitally acquired via HDI Advance structured-light 3D Scanner R3x, with a resolution of 0.25 mm at 600 mm FOV (field of view). The data were processed with FlexScan3D Software (*Figures 6*, *7*, *8d* and *9d*).

### Digital photogrammetry

Several photogrammetric models were obtained using several photos taken with 24 Megapixel Canon EOS 750D (18 mm focal length). The software used to build models is Agisoft PhotoScan Pro (www.agisoft.com). High-resolution Digital Photogrammetry is based on Structure from Motion (SfM) (*Ullman, 1979*) and Multi View Stereo (MVS) (*Seitz et al., 2006*) algorithms and produces high-quality dense point clouds. The accuracy of the obtained models is up to 1 mm for close-range photography. The reconstructed 3D surfaces were then processed in the open-source software Paraview. False colored models with contour lines, highlighting general morphology and differential depth of impression of the traces, were obtained (*Figures 9a*, *12* and *13*).

### Analysis of human footprints

All recognized tracks (107 human traces) were analyzed directly in the field through a morphological approach using available landmarks (*Robbins, 1985*). The differential depth of each individual impression was analyzed directly in the field to infer the complex and multiphase biomechanics. All isolated footprints, and those

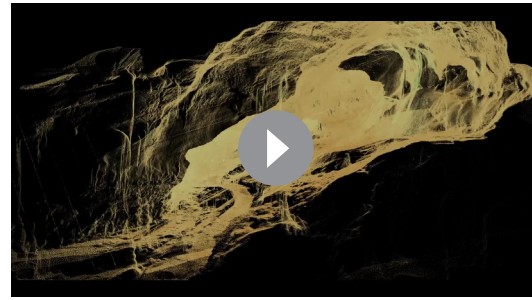

**Video 1.** Virtual exploration of the cave showing the crawling locomotion adopted by the Palaeolithic group to cross the main gallery and to access the innermost rooms of the cave.

DOI: https://doi.org/10.7554/eLife.45204.019

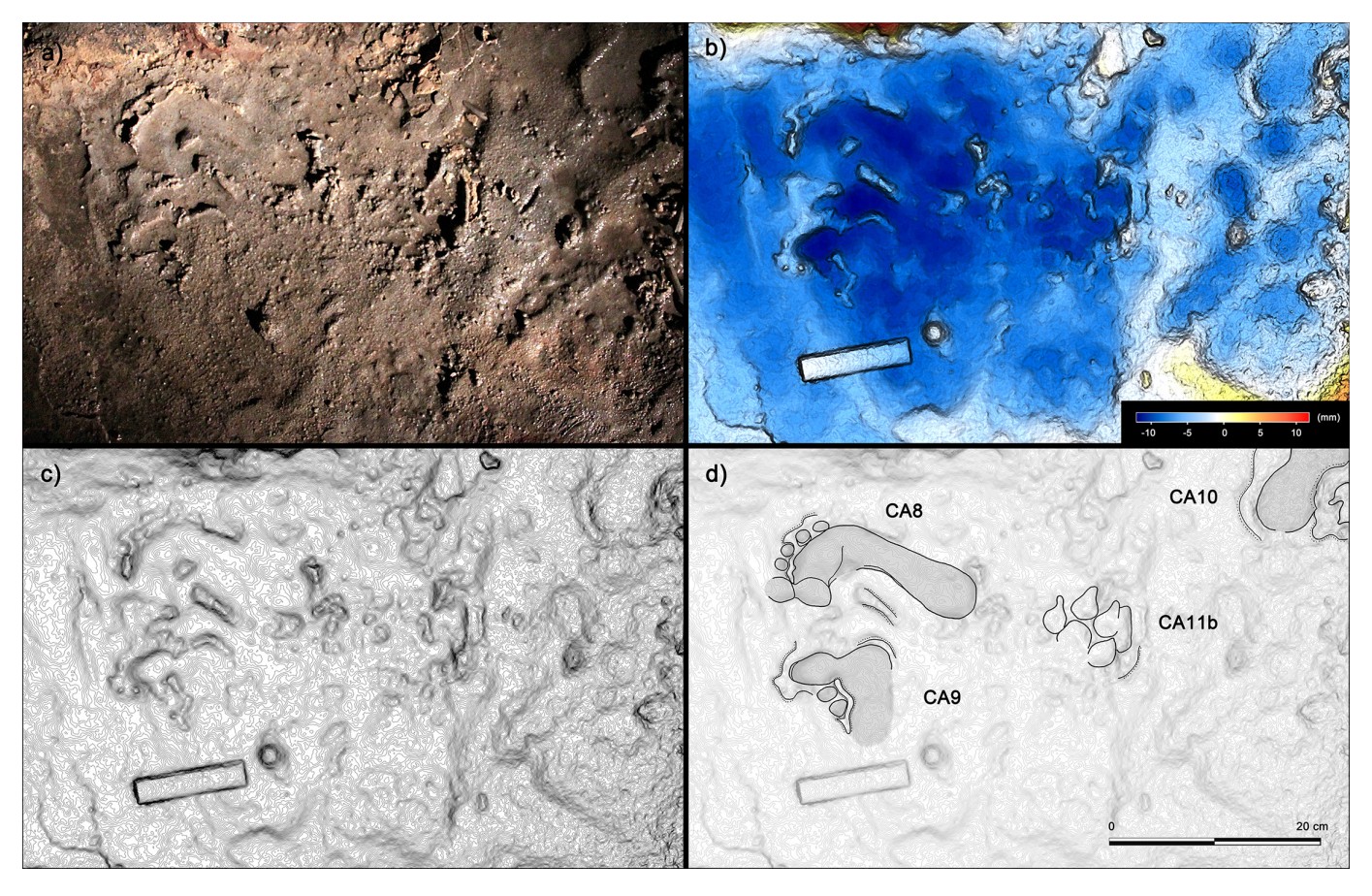

**Figure 13.** Shallow human tracks from the 'upper corridor'. (**a**) Tracks CA8, CA9, CA10 and Ca11b from the sector C of the 'upper corridor' (see *Figure 1* main text). (**b**) Digital terrain model obtained from high-resolution photogrammetry. (**c**) Topographic profile with contour lines, obtained from b. (**d**) Interpretive draw. Tracks were impressed on a hard carbonate substrate covered by a thin muddy deposit, few millimeters in thickness.
DOI: https://doi.org/10.7554/eLife.45204.020

associated with trackways, were drawn in the field on plastic film. All morphological and dimensional data collected in the field were double-checked by using photos and photogrammetric models.

In addition, the original casts of the footprints from the 1950s were also used and analyzed. Their positions in the cave were verified and it was established that all the footprints identified during the early explorations are still visible in situ (*Figure 1*, *Table 4*). Some of them have suffered alteration and loss of detail and others have been partially damaged. For this reason, in some instances, we have integrated morphometric data of the in situ footprints with those of plaster casts. In addition, the following two indices were considered: Footprint index (FI), equal to foot width/foot length x100 and Arch angle (Aa), represented by the angle between the footprint medial border line and the line that connect the most medial point of the footprint metatarsal region and the apex of the concavity of the arch of the footprint (*Clarke, 1933*). In the reconstruction of body dimensions and age, only the foot measurements derived from the better-preserved footprints were used (*Table 3*).

## Principal Component Analysis

The 23 better preserved footprints were subjected to a Principal Component Analysis (PCA) using the software PAST 3.10 (*Hammer et al., 2001*). Homologous points were selected on the footprints for the measurements (*Robbins, 1985*) (*Figure 15*). These included nine anatomical lengths and widths (foot lengths (Dt1-BL, Dt2-BL, Dt3-BL); ball medial length (mtm-BL); ball lateral length (mtl-BL); heel medial length (ccm-BL); heel lateral length (ctul-BL); widths of ball (mtm-horiz.) and heel (ctul-horiz.) (*Table 4*). The raw data were log-transformed before the analysis to fit linear models and

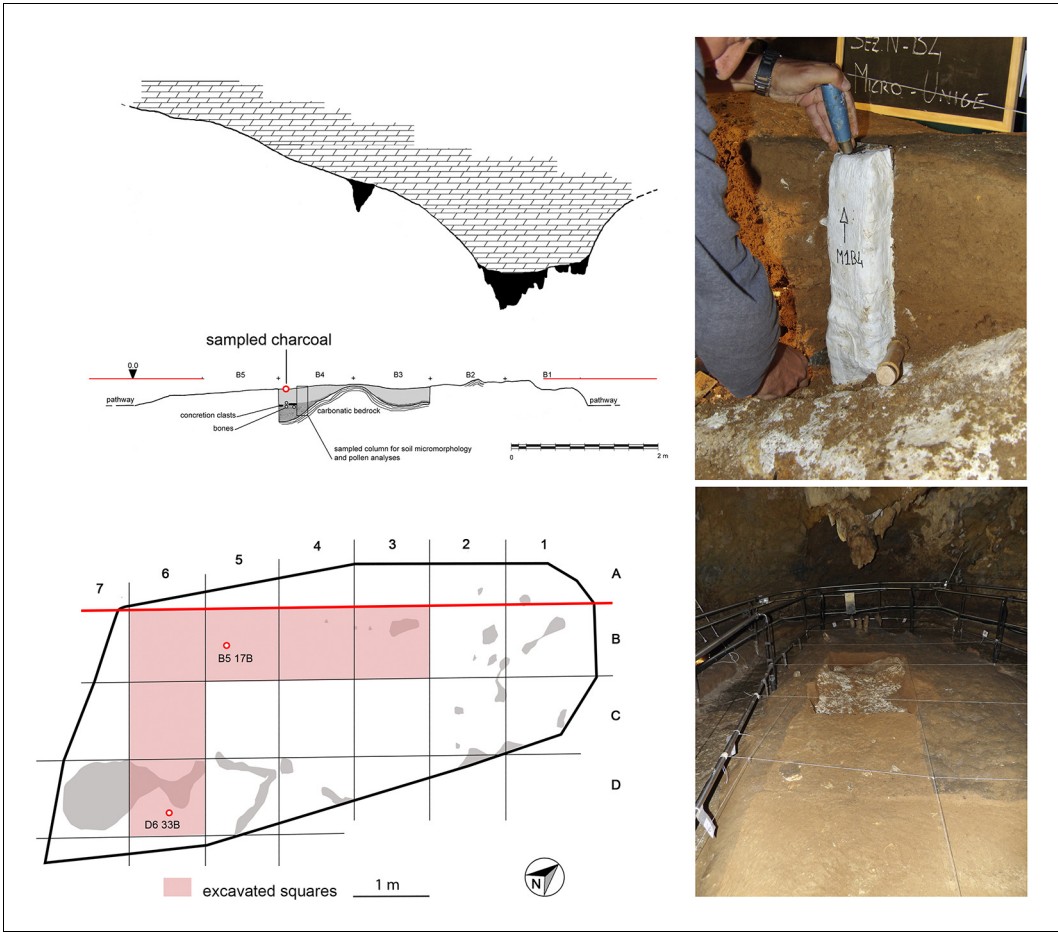

**Figure 14.** Profile and map of the archaeo-paleontological excavations in the Mysteries Hall (left), soil micromorphology sampling and view of the excavations (right). The sampled charcoal for dating are highlighted by a red dot.

DOI: https://doi.org/10.7554/eLife.45204.021

for the correspondence of the log transform to an isometric null hypothesis (*Chinnery, 2004*; *Cheng et al., 2009*; *Romano and Citton, 2015*; *Romano and Citton, 2017*; *Romano, 2017*). Missing entries were treated according to the 'iterative imputation' in PAST 3.10, preferable to the simple 'mean value imputation' (*Hammer, 2013*). The results of the PCA are reported in the scatter plots of *Figure 5a*, whereas the loadings for the first three principal components are provided as supplementary information (and Appendix 1 Tables A, B, *Figure 5—figure supplement 1*).

## Stature

It is possible to estimate stature from foot length (*Robbins, 1985*; *Oberoi et al., 2006*; *Krishan and Sharma, 2007*; *Kanchan et al., 2008*; *Pawar and Pawar, 2012*). Stature varies with race, age, sex, heredity, climate and nutritional status. Based on skeletal evidence, it is thought that the body proportions of terminal Upper Palaeolithic individuals was similar to that of modern humans (*Trinkaus, 1997*; *Ruff et al., 2005*; *Shackelford, 2007*) but the foot length/stature ratio was considered highly uncertain, between 0.15 and 0.16. Consequently, we calculated the foot length/stature ratio based on a sample of terminal Upper Palaeolithic adult individuals (n.8) from the Italian Peninsula (*Corrain, 1977*; *Paoli et al., 1980*; *Formicola et al., 1990*; *Mallegni and Fabbri, 1995*; *Mallegni et al., 2000*). The calculated ratio is found to be 0.1541, which is close to those proposed for modern humans between the XIX and XX centuries (*Robbins, 1985*; *Topinard, 1878*). Stature estimation from the length of long bones is commonly used in forensic medicine. In this study, we used the percutaneous length of the tibia to check the stature of the Morph. 5 as this measurement

**Table 4.** Footprints and relative measures used for the Principal Component Analysis.
Anatomical abbreviations as in Materials and methods Section.

| Footprints | | | Lengths | | | | | | | Widths | |
|---|---|---|---|---|---|---|---|---|---|---|---|
| ID | IN SITU | CAST 1950–51 | Dt1-BL | Dt2-BL | Dt3-BL | Ball medial (mtm-BL) | Ball lateral (mtl-BL) | Heel medial (ccm-BL) | Heel lateral (ctul-BL) | Ball (mtm-horiz) | Heel (ctul-horiz) |
| | | | (cm) | (cm) | (cm) | (cm) | (cm) | (cm) | (cm) | (cm) | (cm) |
| SM3 | X | | 13 | 12.3 | | 10.5 | 9.2 | 2.3 | 2.4 | 5.5 | 4.3 |
| SM4 | X | | 13.5 | 12.5 | 11.5 | 10.2 | 8 | 2.2 | 1.6 | 5.5 | 4.2 |
| SM43 | X | | 13.5 | 13.5 | | 10.5 | 9.4 | 2.2 | 1.8 | 6 | 4.6 |
| SM17 | X | | 14.2 | 13.8 | 13.5 | 10.2 | 8.5 | 2 | 1.8 | 6.5 | 4 |
| SM5 | X | | 17 | 16.8 | 15.5 | 12 | 12 | 2.5 | 2 | 6.5 | 5 |
| SM42 | X | | 17 | 16.8 | | 12.5 | 11.5 | 3.4 | 2.5 | 7 | 5.4 |
| SM26 | X | | 18 | 16.5 | 15 | | | 2 | 3 | | 5 |
| CA8 | X | X | 20.2 | 19.8 | 19.4 | 15 | 13 | 2.6 | 2 | 7.5 | 4.5 |
| C10 | X | | 20.5 | | 19.2 | 15.5 | 12.5 | 2.8 | 2.6 | 7 | 5.6 |
| SM15 | X | | 20.5 | 20.5 | 18.5 | 15.4 | 13 | 4 | 2 | 7 | 6 |
| SM11 | X | | 21 | 20.2 | 19.7 | 15.4 | 14.5 | 3.3 | 2.3 | 7.5 | 6 |
| SM6 | X | | 21 | 21.5 | 20.5 | 17.5 | 15.5 | 4 | 2.2 | 9.2 | 5.7 |
| SM1 | X | X | 21.2 | 21.3 | 20.8 | 17.4 | 14.8 | 3.8 | 2.5 | 8 | 6 |
| C33 | X | X | 22.2 | 21 | 19.5 | 15.5 | 10.7 | 2.5 | 2 | 10 | 5.8 |
| C36 | X | X | 22.7 | 21.7 | 19.5 | | 14.5 | 3 | 2.5 | | 5.5 |
| M21 | | X | | 21.6 | 21.3 | 16.7 | 14.6 | 3.5 | 2.4 | 9.6 | 6.2 |
| CA1 | X | X | 22.4 | 21.8 | 20.8 | 14.8 | 12.8 | 3.1 | | 8 | |
| CA2 | X | | 22.5 | 22 | 21 | 17.3 | 15.2 | 3.5 | 3.5 | 8.8 | 6.4 |
| C61 | X | X | 23 | 21.7 | 20.8 | 16.4 | 14 | 3.8 | 3 | 7.5 | 5.4 |
| C63 | X | X | 23.3 | 22.2 | 20.8 | 17 | 14 | 3.5 | 3.8 | 8.5 | 5.3 |
| C60 | X | X | 25.3 | 24.2 | 22.7 | 18 | 14.8 | 3.5 | 3.3 | 10.5 | 6.4 |
| C37 | X | X | 25.7 | 23.8 | 22.5 | 18.4 | 14.7 | 3.7 | 3.5 | 10.5 | 7 |
| C35B | X | X | 26.2 | 24.8 | 22.8 | 17 | 13.7 | 3.7 | 2.7 | 9.5 | 6.7 |

DOI: https://doi.org/10.7554/eLife.45204.022

is known to have a strong correlation with body height. We used the relation S = 101.85 + 1.81 x PCTL ±3.73 for male and S = 77.86 + 2.36 x PCTL ±2.94 for female (where S = stature and PCTL = Percutaneous tibial length) (*Lemtur et al., 2017*). For a tibial length of 35 cm, the stature of Morph. 5 is 165.2 ± 3.73 cm, which is comparable to the stature assumed from foot length (166.99 ± 2.93 cm).

## Body mass

Body mass estimates were derived from footprint parameters, based on the assumption that human body proportions have been constant through time (*Dingwall et al., 2013*). Regression formulae are based on mature individuals ranging between 154 and 185 cm in stature (weight Kg = 4.71 + (1.82xFL)) (*Dingwall et al., 2013*; *Bavdekar et al., 2006*; *Ashton et al., 2014*) or on children (*Grivas et al., 2008*) with an average height of 147,44 cm (weigth Kg = −71.142 + (5.259xrigthFL). We have used these formulae for the individual taller than 147 cm (*Table 3*), stature: (b) (*Bavdekar et al., 2006*), (c) (*Grivas et al., 2008*). For the three smaller individuals, we used formulae based on extant Caucasian children between 6 and 11 years old (n. 7147) ranging in stature between 118.6 and 145.7 cm (*Malina et al., 1973*) to develop a mathematical relation between foot length and body mass for young individuals. The report is nonlinear and expressed by the formula mass = $2.2897 \, e^{0.126FL}$ (*Citton et al., 2017*). No dataset are available for

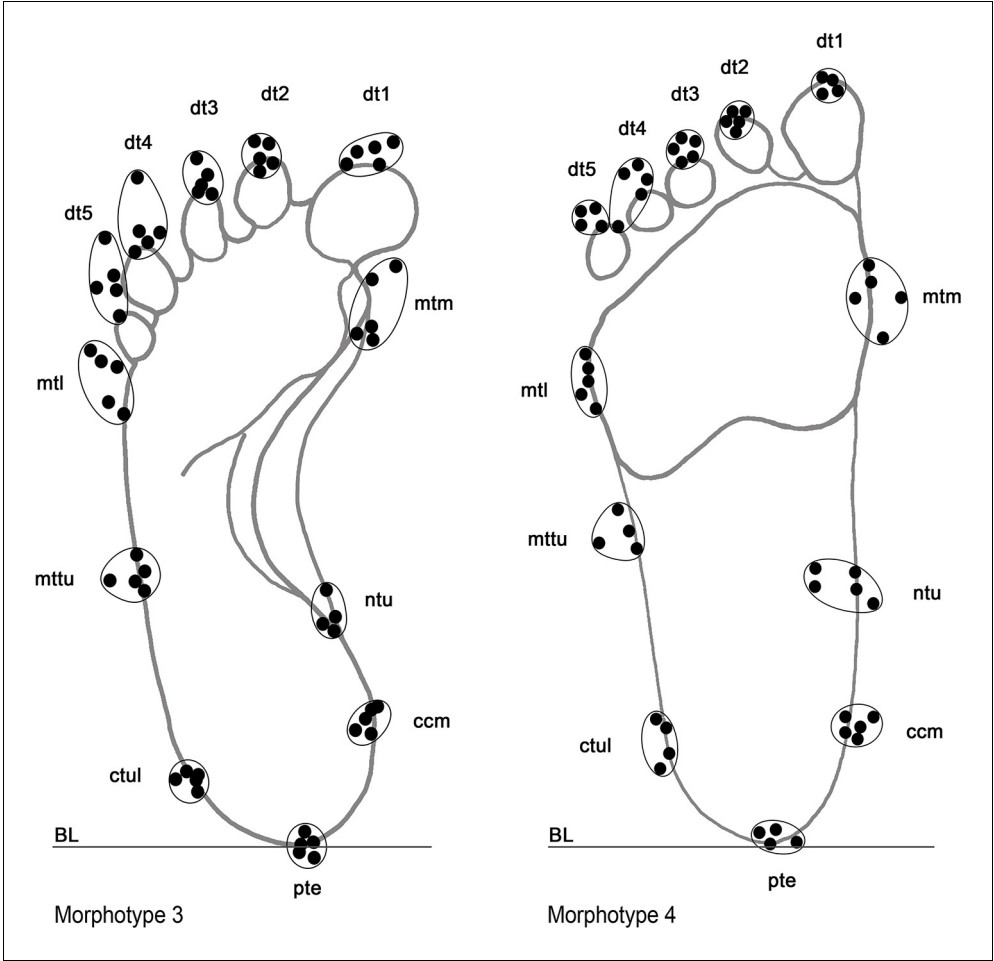

**Figure 15.** Adopted landmarks utilized to perform morphometric analysis, showed in two distinct morphotypes (Morphs 3 and 4) as example. Landmarks in the distal portion of digit traces 4, 5, and in the medial, central and lateral portions of the sole trace were not considered reliable enough for the large variability, higher than the fixed error value (±0.5 cm).

DOI: https://doi.org/10.7554/eLife.45204.023

the smaller individual. We have therefore hypothesized the body mass of MP1 by the trend-line derived from previous formulae.

## Age

Foot length varies according to age and gender. Studies on the relationship size/age of the foot in extant juvenile individuals (*Fryar et al., 2012*; *Müller et al., 2012*) have highlighted that 1-year-old individuals have a foot length equal to 13.07 ± 1.59 cm, reaching 24.4 ± 2.96 cm at the age of 13 years. The age estimation based on growth curves built on extant populations is very similar. However, we must bear in mind that the reference anthropometric data mainly refer to modern well-nourished populations, with a body mass most likely higher at the same age. Anthropometric studies suggest that the morphology of the foot changes and becomes more elongated when the arch stabilizes around the age of six (*Müller et al., 2012*). In extant human populations, from the fifth or sixth years of age, arch angles vary from 21° (3–4 years) to 43° (9–11 years) in young males, and from 26° (3–4 years) to 47° (9–11 years) in young females (*Forriol and Pascual, 1990*). As a result, morphotypes 2 and 3 seem to be similar, likely suggesting a corresponding similarity of age between producers of the two morphotypes. Growth curves based on extant populations with an average height similar of those of the Late Upper Paleolithic provided an estimate of the age of the trackmakers.

For MP5, the wide and stout morphology is here interpreted as an adult stage with the partial collapse of the plantar arch.

## Gender

Sex determination established from the foot has been proposed using Foot index and threshold values. However, this approach is not entirely accepted and some researchers pointed out that the threshold value could vary significantly between populations, thus making it very speculative that gender estimations could be determined from foot morphology (*Walia et al., 2016*). These variations could be due to fact that anatomic structures of the foot manifest ethnic and regional variations owing to climatic factors, physical activities, socio-economic status and nutritional conditions. Despite the method, uncertain arch angle and footprint morphology suggests a possible male as trackmaker of the largest footprint group. No definitive gender can be inferred for Morphotypes 1, 2, 3, 4.

## Acknowledgements

This work was supported by the Soprintendenza Archeologia Belle Arti e Paesaggio per la Città Metropolitana di Genova e le province di Imperia, La Spezia e Savona, Genoa, Italy and the Municipality of Toirano. The funders had no role in study design, data collection and analysis, decision to publish, or preparation of the manuscript. Matthew Bennet provided critical comments to an advanced manuscript draft. Bruce Rubidge and Jonah N Choiniere revised the English language and provided suggestions about the manuscript's structure. We thank Jessica C Thompson; Ignacio Díaz Martínez; Julien Riel-Salvatore for reviewing and Ian Baldwin as the Senior Editor of eLIFE for his cooperation. Part of the research was supported by the National Geographic Early Career Grant to MR. (EC-53477R-18) *"A multidisciplinary approach to a unique human ichnological record from the Grotta della Bàsura (Toirano, Savona Italy)"*.

## Additional information

### Funding

| Funder | Author |
| --- | --- |
| Comune di Toirano | Elisabetta Starnini |
| Università di Genova | Ivano Rellini<br>Marco Firpo<br>Fabio Negrino |
| Museo delle Scienze | Isabella Salvador<br>Marco Avanzini |

The funders provide financial assistance for fieldwork and publication fees.

### Author contributions

Marco Romano, Conceptualization, Data curation, Software, Formal analysis, Investigation, Visualization, Methodology; Paolo Citton, Conceptualization, Investigation, Methodology; Isabella Salvador, Conceptualization, Resources, Data curation, Software, Formal analysis, Validation, Investigation, Visualization, Methodology; Daniele Arobba, Ivano Rellini, Marco Firpo, Formal analysis, Investigation; Fabio Negrino, Validation, Methodology; Marta Zunino, Funding acquisition, Investigation, Project administration; Elisabetta Starnini, Supervision, Funding acquisition, Project administration; Marco Avanzini, Conceptualization, Data curation, Software, Formal analysis, Supervision, Investigation, Visualization, Methodology

### Author ORCIDs

Marco Romano  https://orcid.org/0000-0001-7629-3872
Paolo Citton  http://orcid.org/0000-0002-6503-5541

Isabella Salvador (iD) http://orcid.org/0000-0003-1058-3994
Daniele Arobba (iD) http://orcid.org/0000-0002-6946-7579

**Decision letter and Author response**
Decision letter https://doi.org/10.7554/eLife.45204.030
Author response https://doi.org/10.7554/eLife.45204.031

## Additional files

**Supplementary files**
• Supplementary file 1. Supplementary table with all the tracks analyzed in the study.
DOI: https://doi.org/10.7554/eLife.45204.024

• Transparent reporting form
DOI: https://doi.org/10.7554/eLife.45204.025

**Data availability**
All the data are included in tables within the main text.

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

## Appendix 1

DOI: https://doi.org/10.7554/eLife.45204.026

**Appendix 1—table 1. Scores obtained from the Principal Component Analysis.**

| ID | PC 1 | PC 2 | PC 3 | PC 4 | PC 5 | PC 6 | PC 7 | PC 8 | PC 9 |
|---|---|---|---|---|---|---|---|---|---|
| SM3 | −0.39913 | 0.10426 | −0.064014 | 0.014386 | −0.025072 | 0.016054 | 0.016716 | 0.011934 | 0.0074328 |
| SM4 | −0.50068 | −0.042109 | −0.014907 | 0.032287 | 0.0032614 | −0.029028 | 0.019643 | −0.026998 | 0.0033618 |
| SM43 | −0.40104 | −0.019701 | −0.0054895 | 0.018735 | 0.0016977 | 0.027812 | −0.0033284 | 0.0083841 | −0.0075861 |
| SM17 | −0.42507 | −0.0052715 | 0.061746 | 0.0063252 | −0.038382 | −0.016919 | −0.0164 | 0.0059965 | −0.0012712 |
| SM5 | −0.1976 | −0.035523 | −0.0043672 | −0.03242 | 0.024328 | 0.032498 | −0.025789 | −0.011295 | −0.0026746 |
| SM42 | −0.086943 | 0.013761 | −0.08784 | 0.050378 | −0.009608 | −0.0043334 | −0.025838 | 0.015148 | 0.0033864 |
| SM26 | −0.20626 | 0.16623 | 0.063691 | −0.021075 | 0.045203 | 0.0078808 | −6,97E-02 | −0.013948 | −0.00040047 |
| CA8 | −0.051293 | −0.079409 | 0.054639 | −0.097538 | −0.026619 | −0.014175 | −0.002064 | 0.014186 | 0.00038865 |
| C10 | 0.0058167 | 0.019522 | 0.0058477 | −0.035311 | 0.03647 | −0.00018801 | 0.032056 | 0.019886 | 0.0034661 |
| SM15 | 0.040758 | −0.12904 | −0.095157 | 0.017228 | 0.043206 | −0.034662 | 0.00082729 | −0.005932 | −0.0085001 |
| SM11 | 0.06668 | −0.064129 | −0.032745 | −0.023447 | 0.034482 | 0.026778 | −0.0049565 | 0.0063519 | 0.0068743 |
| SM6 | 0.15624 | −0.12859 | −0.045021 | 0.0020611 | −0.050337 | 0.019214 | 0.001418 | −0.0080637 | −0.0071906 |
| SM1 | 0.14791 | −0.061674 | −0.059992 | −0.0097849 | 0.00034344 | 0.0088016 | 0.019386 | 0.010087 | −0.0078662 |
| C33 | 0.0099173 | −0.09178 | 0.16838 | 0.043167 | 0.0021726 | −0.017624 | 0.012848 | 0.010468 | −0.0049062 |
| C36 | 0.094929 | −0.029812 | 0.032258 | −0.047049 | 0.0025717 | 0.013716 | −0.009842 | −0.02199 | 0.0015418 |
| M21 | 0.16838 | −0.0796 | 0.021892 | 0.019189 | −0.011351 | 0.022302 | −0.0073946 | 0.0015577 | 0.014584 |
| CA1 | 0.16391 | 0.21771 | −0.0025937 | 0.0040729 | 0.0067048 | −0.016137 | −0.017977 | 0.012136 | −0.0084853 |
| CA2 | 0.22089 | 0.072731 | −0.022013 | 0.006969 | −0.0012392 | 0.036927 | 0.012619 | −0.0038898 | −0.0063643 |
| C61 | 0.15271 | 0.023445 | −0.066418 | −0.034022 | −0.0050802 | −0.045788 | −0.0011026 | −0.0017506 | 0.007746 |
| C63 | 0.19883 | 0.1176 | −0.020327 | −0.025171 | −0.043065 | −0.029182 | 0.0070871 | −0.013733 | −0.00144 |
| C60 | 0.28381 | 0.030453 | 0.049268 | 0.0231 | −0.021083 | 0.01058 | −0.0018458 | −0.0051978 | −0.00067736 |
| C37 | 0.3109 | 0.046052 | 0.029704 | 0.054401 | −0.0019844 | 0.016719 | 0.011316 | −0.0085163 | 0.0054307 |
| C35B | 0.24634 | −0.045122 | 0.033459 | 0.033518 | 0.033381 | −0.031244 | −0.01731 | 0.0051775 | 0.0031501 |

DOI: https://doi.org/10.7554/eLife.45204.027

**Appendix 1—table 2. Loadings for each principal components.** a, Dt1-BL; b, Dt2-BL; c, Dt3-BL; d, Ball medial (mtm-BL); e, Ball lateral (mtl-BL); f, Heel medial (ccm-BL); g, Heel lateral (ctul-BL); h, Ball (mtm-horiz); i, Heel (ctul-horiz). Anatomical abbreviations as in Methods section.

| | PC 1 | PC 2 | PC 3 | PC 4 | PC 5 | PC 6 | PC 7 | PC 8 | PC 9 |
|---|---|---|---|---|---|---|---|---|---|
| a | 0.36632 | −0.031724 | 0.28459 | −0.15437 | 0.31322 | −0.38384 | −0.061146 | −0.3656 | 0.61412 |
| b | 0.36721 | −0.086634 | 0.19468 | −0.1893 | 0.19705 | −0.27853 | −0.3069 | −0.13269 | −0.74546 |
| c | 0.34834 | −0.062518 | 0.10263 | −0.17626 | −0.035551 | −0.093333 | −0.12422 | 0.88253 | 0.16995 |
| d | 0.34521 | −0.15 | 0.03307 | −0.16341 | −0.13051 | −0.019176 | 0.88858 | −0.036893 | −0.14898 |
| e | 0.33106 | −0.13532 | −0.24619 | −0.50577 | −0.065819 | 0.67412 | −0.2218 | −0.19439 | 0.099588 |
| f | 0.35243 | −0.24194 | −0.73379 | 0.3425 | −0.19673 | −0.31842 | −0.1176 | −0.070489 | 0.051056 |
| g | 0.32186 | 0.93175 | −0.1145 | 0.050472 | −0.09853 | 0.019641 | 0.025317 | −0.030012 | −0.030921 |
| h | 0.30501 | −0.13213 | 0.50725 | 0.46978 | −0.57082 | 0.21548 | −0.15222 | −0.12269 | 0.031883 |
| i | 0.245 | −0.051134 | −0.0026745 | 0.53499 | 0.68278 | 0.40417 | 0.10113 | 0.098963 | −0.039702 |

DOI: https://doi.org/10.7554/eLife.45204.028

