## [Decision Letter]

Thank you for submitting your article "A multidisciplinary approach to a unique Palaeolithic human ichnological record from Italy (Bàsura Cave)" for consideration by *eLife*. Your article has been reviewed by three peer reviewers, including Jessica C. Thompson as the Reviewing Editor and Reviewer #1, and the evaluation has been overseen by Ian Baldwin as the Senior Editor. The following individuals involved in review of your submission have agreed to reveal their identity: Ignacio Díaz Martínez (Reviewer #2); Julien Riel-Salvatore (Reviewer #3).

The reviewers have discussed the reviews with one another and the Reviewing Editor has drafted this decision to help you prepare a revised submission.

Summary:

This manuscript reports a singular ichnological record left by a group of people who explored the Bàsura Cave around 14 thousand years ago. The authors complete a multidisciplinary study that complements the ichnological one based mainly on sedimentology, geochemistry, archaeobotany, geometric morphometrics, and 3D-modeling. They described footprints, handprints, knee and digit traces, etc. along two corridors and a cave chamber. The authors were able to identify 5 morphotypes of human footprints, which related to five ontogenetic states, two adults, an adolescent, and two children. The data and the methodology that the authors used in this study were high-quality and comprehensive: laser scanner and photogrammetrical 3D models of the cave and the trampling surfaces; new geochronological data; multivariate statistical analysis; biomorphic and biomechanical equations, etc. Therefore, the reconstruction proposed by the authors is based on solid arguments.

This is an excellent, thorough study on important material from an important site. It further uses innovative and cutting-edge methods (notably morphometrics) to reach new, original conclusions about human behavior in the Upper Paleolithic, shedding new light on what is essentially a snapshot of life at that time. The paper makes a real contribution to our understanding of life ca. 14,000 years ago. The combined findings about the number of individuals who visited the cave, their likely ages and the mode of their locomotion during their exploration are major contributions, as is the description of some of the behaviors in Sala dei Misteri. To this we can add the inference about the lack of footwear or clothing over the part of the bodies that left traces in the cave, which is instructive in its own right. The images are appropriate, convincing, and well-executed, and the video offers considerable vividness to their reconstruction. This work should finally give Grotta della Bàsurathe exposure in the international paleoanthropological community which it richly deserves by virtue of its uniqueness and remarkable degree of preservation.

Essential revisions:

1) Details of the dating: What is the evidence that this set of tracks represented a single event, rather than multiple events by different people over several visits? The close 14C dates on charcoal are compelling, but there is little context provided as to where the charcoal was recovered relative to the tracks. In the methods, the exact place from which the samples were derived relative to the marks should be made clear. There are data such as the square number etc. provided for each sample, but without knowing how the excavations and samples were positioned relative to the traces, it is unclear how they relate. Also, where the 14C dates are reported in the body of the manuscript, they should be indicated as being uncalibrated 14C dates BP (right now they don't specify). This becomes confusing where in the text, the authors suggest that the human explored the cave about 14,000 years ago, but the new data of the charcoal remains (Table 1) are around 12,000 years ago.

2) Details and discussion of the canid tracks: What is the significance of the canid tracks? With ongoing interest in the domestication of the dog, and the strong evidence that this process began in the Paleolithic, what are the implications of these tracks being interspersed with human tracks? Are they continuously associated with the human tracks, or do they appear incidental, as do the cave bear tracks? Figure 3 shows some tantalizing patterns, but they are not discussed further. This would be unusual behavior for most canids to enter the dark zone of a deep cave, although denning cannot be ruled out. Were there also canid bones as well as ursids? In the absence of such evidence, the authors should make an effort to establish why they do or do not believe the canid tracks are actually associated with the human entry. This could potentially be one of the more significant findings of the paper, but it is barely mentioned.

3) Details of the methods for trace recording, and which traces still remain in the cave: The methods are not quite detailed enough to establish the basis for the original data collection. Many of the figures describe casts from 1950, and this is noted in the methods, but then there are also parts of the methods that refer to tracing prints in situ. Are all of the traces now gone, or just some of them? How reliable are the casts likely to be? How much of the original traces are likely preserved in the casts? Are the positions of the traces as noted in the cave based only on casts, or on a combination of casts, historical notes, and modern remaining traces? If possible, each print described in Table 4 should have a descriptor saying where in the cave it is from and if it was in situ or a cast.

4) Details of other findings/context from the cave: The manuscript says Bàsura Cave contains some of the most significant human Paleolithic discoveries in Italy, and citations are provided, but a summary of what those discoveries are would be useful. What are they? How old are they? How do they likely relate to the activities recorded in the deeper parts of the cave?

5) Discussion of the symbolic context of the traces: What is the evidence that the finger flutings and finger traces shown in Figure 2 were created in a symbolic context? The presence of humans at the rear of the cave might suggest such behavior, but some finger traces themselves may be accidental, especially if they co-occur with the knee marks and suggest crawling. Perhaps those inferred to be symbolic should be in a separate figure from those inferred to derive from locomotion? The flutings appear significant, but without a bit more description of them it is difficult to discern which ones are actually on the walls/stalactites and which ones are just incidental and/or located on the floor. Also, what is the evidence that clay was collected from the floor? Are there places where this excavation seems to have taken place? The citations direct the reader to some of the primary descriptions, but a summary would be invaluable here.

6) Restriction of discussion to the evidence: In the discussion, there are some areas when the manuscript over-interprets the data. How do the trackways cause the inference that body sizes were slender and muscular? Is this based mostly on body mass estimates? The knee impressions do not seem to speak to this, since they are a part of the anatomy that has protruding bones and ligaments when kneeling in spite of the degree of slenderness and muscularity. Also, the ensuing interpretation that this was a hyper-active teenage explorer is an over-interpretation of the data and should be removed. It distracts from the more solid findings.

7) Revision of the discussion about the role of children: Another important conclusion of this study is that even very young children were included in potentially dangerous activities at the time. In that light, however, the authors should nuance their discussion of local comparative data from the Arene Candide and Arma Veirana, because all these show is that children were considered important members of their groups rather than saying anything about how they participated in their societies. In this sense, perhaps a better indirect referent for the social place of children in Epigravettian groups would be the small size of the pebbles argued to have been broken as part of mortuary behavior at the Arene Candide argued by Gravel-Miguel. I think this speaks more about the involvement of children as opposed to simply mentioning they were interred, which shows they were there but says little about their role.

8) Ichnological terminology: The authors used footprint, traces, tracks, etc. However, for all the ichnological record sensu lato, trace is the best option. If the trace is impressed by the foot, footprint, if it is impressed by the hand, handprint, etc.

9) Copy-editing: The paper needs to be thoroughly edited for the quality of its English. There are currently many odd turns of phrase and instances of transliteration from Italian to English that can be a bit jarring and impede smooth reading.

---

## [Author Response]

Essential revisions:1) Details of the dating: What is the evidence that this set of tracks represented a single event, rather than multiple events by different people over several visits? The close 14C dates on charcoal are compelling, but there is little context provided as to where the charcoal was recovered relative to the tracks. In the methods, the exact place from which the samples were derived relative to the marks should be made clear. There are data such as the square number etc. provided for each sample, but without knowing how the excavations and samples were positioned relative to the traces, it is unclear how they relate. Also, where the 14C dates are reported in the body of the manuscript, they should be indicated as being uncalibrated 14C dates BP (right now they don't specify). This becomes confusing where in the text, the authors suggest that the human explored the cave about 14,000 years ago, but the new data of the charcoal remains (Table 1) are around 12,000 years ago.

In Figure 1 we have added the original position of the charcoal sample used for dating. The charcoal remains are those of the bundles originally utilized to illuminate the cave and preserved on the trampling palaeosurface of the “Sala dei Misteri”.

We add a new paragraph in the text highlighting all the evidences that make a single exploratory event for the cave more parsimonious, and the only one possible on the base of the available data. In particular, the short radiometric interval documented by both previous radiometric data and those derived from the latest research here presented (Table 1), together with the superimpositions between footprints allowing to decipher the timing of each singular impression, suggest that all individuals entered the cave in a single episode. In particular, the group of footprints C34, C35b, C36, C35 in the terminal part of the corridor of the footprints provides the timing of the impression: C35 (morph4) is superimposed on C36 (morph3) and the latter by C35 (morph4) (Figure 8). In the initial part of the same corridor it is observed that C63 (morph4) is superimposed by C64 (morph3) proving that morphs 3 and 4 are certainly contemporary. It is also difficult to assume that morphs 1 and 2 may have entered the deepest part of the cave without the presence of some older individual: if the older individuals who were leading the exploration were either morphs 3 or 4, the group consequently increases to include at least 4 individuals. No relationship can be established for the larger individual; however, as reported below, the fact that the footprints of this individual are regularly trampled and followed exactly on its path by all the other individuals, suggests that the larger one performed very likely the function of leader during a synchronous exploration. Therefore, the set of available evidences, and in particular the complex interference between the studied footprints, strongly suggests that a single exploration event by a heterogeneous restricted group of five individuals represents the most parsimonious and better supported hypothesis.

2) Details and discussion of the canid tracks: What is the significance of the canid tracks? With ongoing interest in the domestication of the dog, and the strong evidence that this process began in the Paleolithic, what are the implications of these tracks being interspersed with human tracks? Are they continuously associated with the human tracks, or do they appear incidental, as do the cave bear tracks? Figure 3 shows some tantalizing patterns, but they are not discussed further. This would be unusual behavior for most canids to enter the dark zone of a deep cave, although denning cannot be ruled out. Were there also canid bones as well as ursids? In the absence of such evidence, the authors should make an effort to establish why they do or do not believe the canid tracks are actually associated with the human entry. This could potentially be one of the more significant findings of the paper, but it is barely mentioned.

In fact, considering the importance of dog’s footprints as evidence for the domestication of the dog in the Paleolithic, this material is currently still under study and the object of a separate dedicated project; the latter includes the analysis of numerous trackways of extant dogs, hyenas and wolves to test whether the Bàsura dog footprints can be morphometrically referred to a single individual, and a detailed study of the interferences with the human footprints, to verify the real contemporaneity of frequentation of the hypogeal environment by dogs and humans. In any case, we included a short sentence in the new manuscript, along with some related bibliography, highlighting the possible importance of this material in the framework of dog domestication in the Paleolithic, stressing that this material is the subject of still ongoing study.

3) Details of the methods for trace recording, and which traces still remain in the cave: The methods are not quite detailed enough to establish the basis for the original data collection. Many of the figures describe casts from 1950, and this is noted in the methods, but then there are also parts of the methods that refer to tracing prints in situ. Are all of the traces now gone, or just some of them? How reliable are the casts likely to be? How much of the original traces are likely preserved in the casts? Are the positions of the traces as noted in the cave based only on casts, or on a combination of casts, historical notes, and modern remaining traces? If possible, each print described in Table 4 should have a descriptor saying where in the cave it is from and if it was in situ or a cast.

We add a paragraph in the Material and methods section stressing that in our direct analysis in the site we have also considered and analyzed the original casts from the 50s. Their position has been identified in the cave and it has been verified that all the footprints identified during the first explorations are still visible in situ (Figure 1). Some of them have suffered alterations and loss of details, others have been partially damaged. For this reason, in some cases we have integrated the morphometric data of the original in situ with those of plaster casts.

4) Details of other findings/context from the cave: The manuscript says Bàsura Cave contains some of the most significant human Paleolithic discoveries in Italy, and citations are provided, but a summary of what those discoveries are would be useful. What are they? How old are they? How do they likely relate to the activities recorded in the deeper parts of the cave?

In this regard we add some additional paragraphs in the Introduction, highlighting chronologically the first studies and published contributions to the Bàsura Cave immediately after the discovery. The historical overview, reported with sufficient detail, but without overloading to much the new version of the text, highlights the major contributions made by Tongiorgi, Lambroglia, Chiappella, Blanc and Pale, with the original interpretation of the human footprints, number of individuals and ritual interpretation of some traces and evidences.

5) Discussion of the symbolic context of the traces: What is the evidence that the finger flutings and finger traces shown in Figure 2 were created in a symbolic context? The presence of humans at the rear of the cave might suggest such behavior, but some finger traces themselves may be accidental, especially if they co-occur with the knee marks and suggest crawling. Perhaps those inferred to be symbolic should be in a separate figure from those inferred to derive from locomotion? The flutings appear significant, but without a bit more description of them it is difficult to discern which ones are actually on the walls/stalactites and which ones are just incidental and/or located on the floor. Also, what is the evidence that clay was collected from the floor? Are there places where this excavation seems to have taken place? The citations direct the reader to some of the primary descriptions, but a summary would be invaluable here.

Also, in this case, we are conducting a work totally dedicated to the analysis of the authors and age of the finger flutings, with 3D reconstruction of the “zoomorphic stalagmite” and identification of the possible source area from which the clay material was excavated. A figure has been added where a sample of involuntary types from those created in a possible symbolic context is differentiated.

6) Restriction of discussion to the evidence: In the discussion, there are some areas when the manuscript over-interprets the data. How do the trackways cause the inference that body sizes were slender and muscular? Is this based mostly on body mass estimates? The knee impressions do not seem to speak to this, since they are a part of the anatomy that has protruding bones and ligaments when kneeling in spite of the degree of slenderness and muscularity. Also, the ensuing interpretation that this was a hyper-active teenage explorer is an over-interpretation of the data and should be removed. It distracts from the more solid findings.

Yes, the slender and muscular body have been inferred on the base of body size estimate starting from footprint parameters. In this regard we add the following sentence to the new version of the text: “Body mass estimates derived from footprints parameters suggest slender and muscular body size for all the trackmakers. Arch angle and footprint morphology suggests a possible male as trackmaker of the largest footprint group. For the morphotypes 1, 2, 3, 4 any inference of gender result possible although the presence of almost a female (morph.4?) it seems probable.”

As properly suggested, the mention to ‘hyper-active teenage explorer’ has been completely removed from the text.

7) Revision of the discussion about the role of children: Another important conclusion of this study is that even very young children were included in potentially dangerous activities at the time. In that light, however, the authors should nuance their discussion of local comparative data from the Arene Candide and Arma Veirana, because all these show is that children were considered important members of their groups rather than saying anything about how they participated in their societies. In this sense, perhaps a better indirect referent for the social place of children in Epigravettian groups would be the small size of the pebbles argued to have been broken as part of mortuary behavior at the Arene Candide argued by Gravel-Miguel. I think this speaks more about the involvement of children as opposed to simply mentioning they were interred, which shows they were there but says little about their role.

In this respect, we have underlined the fact that not only some young individuals were buried, but that they were instead the object of particular attention. Indeed, we have added a mention to the grave goods found in association with the newborn from Arma Veirana and references are made to the sepulchral rites brought to light at the Arene Candide cemetery, along with the related bibliography. In our opinion, the constant association of adults and young individuals both in the burials and in external activities underlines the complex organization of Epigravettian and Mesolithic social groups, where even very young individuals, such as a newborn, could be fully included in the band structure both sharing their own responsibilities with adults and making themself object of elaborate funeral rites.

8) Ichnological terminology: The authors used footprint, traces, tracks, etc. However, for all the ichnological record sensu lato, trace is best option. If the trace is impressed by the foot, footprint, if it is impressed by the hand, handprint, etc.

We prefer to use the general term ‘trace’ sensu lato, since the material is not represented only by footprints and handprint (e.g. traces of knee and proximal shin).

9) Copy-editing: The paper needs to be thoroughly edited for the quality of its English. There are currently many odd turns of phrase and instances of transliteration from Italian to English that can be a bit jarring and impede smooth reading.

The new version of the manuscript was kindly reviewed carefully by the native speaker Professor Bruce Rubidge, consistently improving the quality of the English.